# TRAINING VERIFIABLY ROBUST AGENTS USING SET-BASED REINFORCEMENT LEARNING

## ABSTRACT

Deep reinforcement learning uses neural networks to solve complex control tasks. However, neural networks are sensitive to input perturbations, which makes their deployment in safety-critical environments challenging and thus their formal verification necessary. This work lifts recent results from formal verification of neural networks to reinforcement learning in continuous state and action spaces. While previous work mainly focuses on adversarial attacks for robust reinforcement learning, we augment reinforcement learning with set-based computing: We enclose all possible outputs for a set of perturbed inputs and compute a gradient set for training, i.e., each possible output has a different gradient. Thereby, we control the size of the propagated sets, yielding favorable worst-case bounds for actions and value functions that enable formal verification across different verification frameworks for up to 9 times larger input perturbations. Our work addresses the gap between state-of-the-art adversarial training methods and formal verification to train verifiably robust agents, making them applicable in safety-critical environments.

## 1 INTRODUCTION

In recent years, deep reinforcement learning using neural networks has significantly improved solving complex control tasks (Mnih et al., 2015; OpenAI et al., 2020; Lillicrap et al., 2016). In many control tasks, state-action spaces are continuous, high-dimensional, and influenced by inherent system uncertainties, modeling errors, and sensor noise (Kober et al., 2013). However, such uncertainties are challenging for reinforcement learning when parameterizing policies as neural networks, which are sensitive to small input perturbations (Szegedy et al., 2014). This may lead to instabilities and safety violations of the controlled system: Fig. 1(a) shows an example of a navigation task where small input perturbations lead to trajectories that enter an unsafe set. Robustness to such perturbations is therefore essential. For deployment in safety-critical applications, it is additionally necessary to demonstrate safety and guaranteed worst-case performance. This necessitates formal verifiability, which is not inherently guaranteed by current state-of-the-art robust reinforcement learning methods.

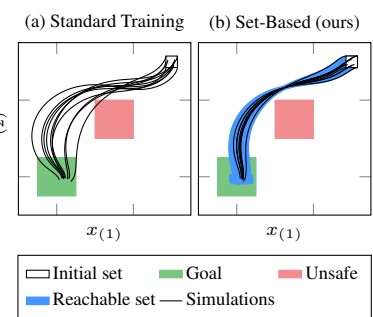

Figure 1: Comparison of standard and our set-based reinforcement learning on a navigation task. (a) Trajectories of the standard agent intersect with the obstacle. (b) We can formally verify our robust agent.[1]

**Related works** on robust reinforcement learning (Zhang et al., 2021a; Huang et al., 2017; Deshpande et al., 2021; Mandlekar et al., 2017; Lütjens et al., 2020; Zhang et al., 2021b) propose a competitive framework with an adversary (Moos et al., 2022; Pinto et al., 2017): In observation-robust algorithms, the adversary exploits the sensitivity of neural networks to choose a worst-case observation for the policy (Moos et al., 2022). Computing the worst-case observation is often intractable (Madry et al., 2018). Consequently, a variety of naive, gradient-based (Pattanaik et al., 2018; Mandlekar et al., 2017;

---

[1]Code: `https://anonymous.4open.science/r/RobustSetBasedRL`
   Video: `https://t1p.de/zjayi`

Huang et al., 2017), learning-based (Zhang et al., 2021a), and convex relaxation methods (Zhang et al., 2020), have been proposed to approximate adversarial observations. For instance, gradient-based methods typically employ the Fast Gradient Sign Method (FGSM) to approximate the most adversarial input (Goodfellow et al., 2015). In contrast, our method does not rely on a single adversarial instance. Instead, it updates the agent parameters using the entire set of admissible perturbations, thereby improving robustness and enabling stronger formal guarantees of verifiability.

**Safe deployment** in safety-critical environments requires more than empirical robustness, it demands formal guarantees. While prior work often relies on adversarial attacks or randomized smoothing (Wu et al., 2022) to determine a probabilistic upper bound on the worst-case performance, these approaches do not provide a provable lower bound. In contrast, our work focuses on formal verification of neural network-based control policies. Recent advances by Manzanas Lopez et al. (2023); Brix et al. (2023) make it possible to verify entire neural network control systems: This is often achieved by (i) modeling the disturbed state of the system as a continuous set, (ii) computing the corresponding output set of the neural networks, and (iii) enclosing the dynamics of the environment over time using reachability analysis. If the obtained reachable set does not violate specifications, the neural network control system is verified as shown in Fig. 1(b).

For the training of robust neural networks, a recent work proposes set-based training: For a set of possible inputs, the set of possible outputs is enclosed and the neural network is trained with a gradient set (Koller et al., 2025), i.e., each possible output has a different gradient. By picking gradients that point toward the center of the output set, the size of the output sets can be controlled. Thereby, the trained neural network is more robust and easier to formally verify with set-based verification algorithms, because smaller propagated sets reduce the conservatism of the verification algorithm. In this work, we lift set-based training to reinforcement learning. Our main contributions are:

(i) A novel set-based reinforcement learning algorithm that, for the first time, computes a gradient set, which contains a different gradient for each possible output given input perturbations. Our algorithm trains agents that are provably more robust and formally verifiable across different available reachability-analysis frameworks.

(ii) A rigorous analysis of the underlying set propagation to derive a novel set-based loss function for regression tasks, which is used to compute gradient sets that optimize over entire output sets and thus achieve formal verifiability of the trained agents given input perturbations.

(iii) An extensive evaluation including a comparison with state-of-the-art adversarial training algorithms and an ablation study to justify our design choices.

## 2 PRELIMINARIES

### 2.1 NOTATION

We write vectors as lowercase letters, matrices as uppercase letters, sets as calligraphic letters, and probability distributions as script font letters. The $i$-th entry of $v \in \mathbb{R}^n$ is written as $v_{(i)}$. The entry in the $i$-th row and $j$-th column of a matrix is $M_{(i,j)}$; $M_{(i,\cdot)}$ is the $i$-th row, and $M_{(\cdot,j)}$ the $j$-th column. The horizontal concatenation of matrices $A \in \mathbb{R}^{n \times m}$ and $B \in \mathbb{R}^{n \times p}$ is denoted by $[A\ B]$. The identity matrix is denoted by $I_n \in \mathbb{R}^{n \times n}$, and the vector containing only ones or zeros is denoted by $\mathbf{1}$ or $\mathbf{0}$. The set of natural numbers up to $n \in \mathbb{N}$ is written as $[n] = \{1, 2, \ldots, n\} \subset \mathbb{N}$. We denote a multidimensional interval by $\mathcal{I} = [l, u] = \{x \in \mathbb{R}^n \mid \forall i \in [n]\colon l_{(i)} \leq x_{(i)} \leq u_{(i)}\}$. The gradient of a function $f$ w.r.t. a variable $x$ is denoted by $\nabla_x f(x, \cdot)$. The operator $\mathrm{diag} : \mathbb{R}^n \to \mathbb{R}^{n \times n}$ returns a diagonal matrix with the vector elements on its diagonal. The expected value of a random variable $x$ under condition $y \sim \mathscr{Y}$ is $\mathbb{E}_{y \sim \mathscr{Y}}[x(y)]$.

### 2.2 NEURAL NETWORKS

A feed-forward neural network $N_\theta : \mathbb{R}^{n_0} \to \mathbb{R}^{n_\kappa}$ with learnable parameters $\theta$ consists of $\kappa \in \mathbb{N}$ alternating linear and activation layers, where the $k$-th layer has $n_k \in \mathbb{N}$ output neurons. The output $\hat{y} = N_\theta(x) \in \mathbb{R}^{n_\kappa}$ is computed by propagating an input $x \in \mathbb{R}^{n_0}$ through all layers.

**Definition 2.1** (Neural Network, (Bishop & Nasrabadi, 2006, Sec. 5.1)). *Given a neural network $N_\theta$ and an input $x \in \mathbb{R}^{n_0}$, the output $\hat{y} = N_\theta(x) \in \mathbb{R}^{n_\kappa}$ is given by*

$$h_0 = x, \quad h_k = L_k(h_{k-1}) = \begin{cases} W_k\, h_{k-1} + b_k & \text{if } k\text{-th layer is linear,} \\ \sigma_k(h_{k-1}) & \text{otherwise,} \end{cases} \quad \text{for } k \in [\kappa], \quad \hat{y} = h_\kappa,$$

*with weights $W_k \in \mathbb{R}^{n_k \times n_{k-1}}$, biases $b_k \in \mathbb{R}^{n_k}$, and elementwise activation functions $\sigma_k(\cdot)$.*

## 2.3 DEEP DETERMINISTIC POLICY GRADIENT

We focus on continuous control tasks with a multidimensional state and action space $\mathcal{S}, \mathcal{A}$ (Januszewski et al., 2021; Recht, 2019). Our set-based reinforcement learning approach is based on the deep deterministic policy gradient algorithm (DDPG) (Lillicrap et al., 2016) which consists of an actor $\mu_\phi \colon \mathcal{S} \to \mathcal{A}$ with parameters $\phi$ and a critic $Q_\theta \colon \mathcal{S} \times \mathcal{A} \to \mathbb{R}$ with parameters $\theta$. Starting from an initial state $s_0$, the actor observes the state $s_t \in \mathcal{S}$ at time step $t$ and returns an action $a_t = \mu_\phi(s_t)$, which controls the system until the next time step $t + 1$. Using a reward function $r \colon \mathcal{S} \times \mathcal{A} \to \mathbb{R}$ and the

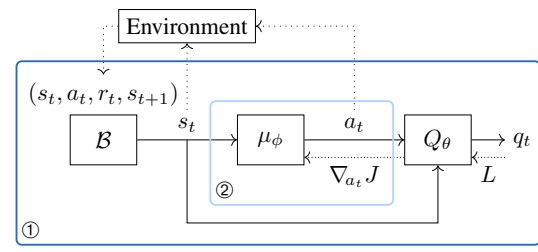

Figure 2: Illustration of the structure of the deep deterministic policy gradient algorithm; ① and ② show the components that are augmented by our set-based training (introduced in Sec. 3).

state transition probabilities $p(s_{t+1}|s_t, a_t)$, the environment returns a reward $r_t$ and its next state $s_{t+1}$ (Fig. 2). These transitions $(s_t, a_t, r_t, s_{t+1})$ are stored in a buffer $\mathcal{B}$ (Fig. 2). The training objective is to find the policy that maximizes the discounted cumulative reward (Silver et al., 2014, Eq. 8):

$$\max_\phi J(\phi, \rho^\mu) = \max_\phi \mathbb{E}_{s_t \sim \rho^\mu} \left[ \sum_{t=0}^\infty \gamma^t\, r(s_t, \mu_\phi(s_t)) \right], \tag{1}$$

with discount factor $\gamma \in [0, 1]$ and discounted state visitation distribution $\rho^\mu$ for policy $\mu$ (Lillicrap et al., 2016, Sec. 2). The critic $Q_\theta$ approximates the expected total discounted reward for action $a_t$ in state $s_t$ (Lillicrap et al., 2016, Eq. 3):

$$Q_\theta(s_t, a_t) = r(s_t, a_t) + \gamma\, \mathbb{E}_{s_{t+1} \sim \rho^\mu}[Q_\theta(s_{t+1}, \mu_\phi(s_{t+1}))]. \tag{2}$$

The critic is trained off-policy with a stochastic policy $\beta$ (Lillicrap et al., 2016, Eq. 4) and targets $y_t$ (Lillicrap et al., 2016, Eq. 5) using the following bootstrapped Q-iterations:

$$L(\theta) = \mathbb{E}_{s_t \sim \rho^\beta, a_t \sim \beta}\big[(Q_\theta(s_t, a_t) - y_t)^2\big], \quad y_t = r(s_t, a_t) + \gamma\, Q_\theta(s_{t+1}, \mu_\phi(s_{t+1})). \tag{3}$$

The actor is trained using the policy gradient (Lillicrap et al., 2016, Eq. 6):

$$\nabla_\phi J(\phi, \rho^\beta) \approx \mathbb{E}_{s_t \sim \rho^\beta} \left[ \nabla_{a_t} Q_\theta(s_t, a_t)\big|_{a_t = \mu_\phi(s_t)} \nabla_\phi \mu_\phi(s_t) \right]. \tag{4}$$

## 2.4 SET-BASED COMPUTATIONS

We model uncertainties using zonotopes, a continuous set representation, due to their favorable computational complexity, propagating the sets of inputs and gradients through the neural networks:

**Definition 2.2** (Zonotope (Girard, 2005)). *Given center $c \in \mathbb{R}^n$ and generators $G \in \mathbb{R}^{n \times q}$, we define*
$$\mathcal{Z} = \langle c, G \rangle_Z = \{c + G\,\beta \mid \beta \in [-1, 1]^q\} \subset \mathbb{R}^n.$$

The affine map $x \mapsto A\,x + b$ of a zonotope $\mathcal{Z} = \langle c, G \rangle_Z$ is computed by (Althoff, 2010, Sec. 2.4)
$$A\,\mathcal{Z} + b = \{A\,x + b \mid x \in \mathcal{Z}\} = \langle A\,c + b, A\,G \rangle_Z. \tag{5}$$

The enclosing interval of zonotope $\mathcal{Z} = \langle c, G \rangle_Z$, i.e., $\mathcal{Z} \subseteq [l, u]$ and its diameter $\mathrm{dia}(\mathcal{Z})$, are computed by (Althoff, 2010, Prop. 2.2)
$$l = c - |G|\,\mathbf{1}, \quad u = c + |G|\,\mathbf{1}, \quad \mathrm{dia}(\mathcal{Z}) := u - l = 2\,|G|\,\mathbf{1}. \tag{6}$$

For some derivations, we write $\mathrm{lnDia}(G) := \ln(2\,|G|\mathbf{1})$ and $\mathrm{lnDia}'(G) := \nabla_G \mathrm{lnDia}(G) = \mathrm{diag}(|G|\mathbf{1})^{-1}\,\mathrm{sign}\,G$ to avoid clutter, $\mathrm{sign}(G)$ returns the sign of each entry of matrix $G$.

The Minkowski sum of a zonotope $\mathcal{Z} = \langle c, G \rangle_Z$ and an interval $\mathcal{I} = [l, u]$ is computed by (Althoff, 2010, Prop. 2.1 and Sec. 2.4):
$$\mathcal{Z} \oplus \mathcal{I} = \{x_1 + x_2 \mid x_1 \in \mathcal{Z}, x_2 \in \mathcal{I}\} = \langle c + \tfrac{1}{2}(u + l), [G\,\mathrm{diag}(\tfrac{1}{2}(u - l))] \rangle_Z. \tag{7}$$

## 2.5 SET PROPAGATION THROUGH NEURAL NETWORKS

Computing the exact output set $\mathcal{Y}^* = N_\theta(\mathcal{X})$ of a neural network for a given input set $\mathcal{X} \subset \mathbb{R}^{n_0}$ is computationally hard, i.e., for neural networks with $\mathrm{ReLU}$-activation it is NP-hard (Katz et al., 2017). Thus, an enclosure of the output set $\mathcal{Y} \supseteq \mathcal{Y}^*$ is computed by conservatively propagating the input set through the layers of the neural network:

**Proposition 2.3** (Neural Network Set Propagation (Singh et al., 2018)). *Given an input set $\mathcal{X}$, the output set of a neural network can be enclosed as:*

$$\mathcal{H}_0 = \mathcal{X}, \quad \mathcal{H}_k = \texttt{enclose}(L_k, \mathcal{H}_{k-1}) \quad \text{for } k \in [\kappa], \quad \mathcal{Y}^* \subseteq \mathcal{Y} = \mathcal{H}_\kappa.$$

The operation $\texttt{enclose}(L_k, \mathcal{H}_{k-1})$ encloses the output set of the $k$-th layer given the input set $\mathcal{H}_{k-1}$. If the $k$-th layer is linear (Def. 2.1), the affine map (5) is applied:

$$\texttt{enclose}(L_k, \mathcal{H}_{k-1}) = W_k\, \mathcal{H}_{k-1} + b_k. \tag{8}$$

The output set of an activation layer is enclosed by approximating its activation function element-wise with a linear function with slope $m_k \in \mathbb{R}^{n_k}$ and approximation error $[\underline{d}_k, \overline{d}_k]$ Koller et al. (2025):

$$\texttt{enclose}(L_k, \mathcal{H}_{k-1}) = \mathrm{diag}(m_k)\, \mathcal{H}_{k-1} \oplus [\underline{d}_k, \overline{d}_k], \quad m_k = \frac{\sigma_k(u_{k-1}) - \sigma_k(l_{k-1})}{u_{k-1} - l_{k-1}}. \tag{9}$$

Intuitively, we can thereby compute the output of the neural network for an infinite number of inputs in a single forward pass, thereby enabling us to adapt learning to account for worst-case outcomes.

## 2.6 PROBLEM STATEMENT

We are given a reinforcement learning task with uncertain initial states $s_0 \in \mathcal{S}_0 \subset \mathbb{R}^n$. Moreover, we model state uncertainties with an $\ell_\infty$-ball of radius $\epsilon \in \mathbb{R}_+$. The set of all perturbations is $\mathcal{V}_\epsilon^\infty := \{\nu \colon \mathbb{R}^n \times \mathbb{R} \to \mathbb{R}^n \mid \forall s_t \in \mathbb{R}^n, \forall t \in \mathbb{R}_+ \colon \|\nu(s_t, t) - s_t\|_\infty \leq \epsilon\}$. Our goal is to use set-based training (Koller et al., 2025) to robustly maximize the reward under adversarial perturbations

$$\max_\phi \min_{\nu \in \mathcal{V}_\epsilon^\infty} J(\phi, \rho^{\mu \circ \nu}), \tag{10}$$

where $\rho^{\mu \circ \nu}$ denotes the state visitation distribution perturbed with adversary $\nu \in \mathcal{V}_\epsilon^\infty$. Further, let $\mathcal{R}$ enclose all reachable states for the time interval $[0, t_{\text{end}}]$. We derive a verified performance as a lower bound on the reward by evaluating the worst trajectory from $\mathcal{R}$.

## 3 SET-BASED REINFORCEMENT LEARNING

We augment actor-critic reinforcement learning with set-based training using a gradient set, i.e., each possible output has a different gradient. Therefore, we propagate sets through the actor and the critic and train both, actor and critic, with gradient sets (SA-SC), which corresponds to a set-based evaluation of ① in Fig. 2. To this end, we extend the critic loss (3), policy gradient (4), and the replay buffer to sets. The main steps are given in Alg. 1.

Given $s_t$, we enclose all possible adversaries:

$$\mathcal{S}_t = \{\nu(s_t, t) \mid \nu \in \mathcal{V}_\epsilon^\infty\} = \langle s_t, \epsilon\, I \rangle_Z. \tag{11}$$

We enclose the set of possible actions by propagating $\mathcal{S}_t$ through the actor (Prop. 2.3):

$$\mathcal{A}_t = \langle c_{\mathcal{A}_t}, G_{\mathcal{A}_t} \rangle_Z = \texttt{enclose}(\mu_\phi, \mathcal{S}_t). \tag{12}$$

For the off-policy training of the critic, we perturb the set of actions with random exploration noise $e_t$ (Lillicrap et al., 2016, Eq. 7): $\tilde{\mathcal{A}}_t := \mathcal{A}_t + e_t$, and compute the corresponding set of critic outputs:

---

**Algorithm 1** Set-based reinforcement learning.

1: Randomly initialize $Q_\theta$, $\mu_\phi$ with $\theta, \phi$
2: Initialize replay buffer $\mathcal{B}$
3: **for** episode $= 1, \dots,$ maxEpisodes **do**
4:      Get initial observation $s_0$
5:      **for** $t = 0, 1, \dots,$ maxSteps **do**
6:          $\mathcal{S}_t \leftarrow \langle s_t, \epsilon I \rangle_Z$ {perturb state Eq. (11)}
7:          $\mathcal{A}_t \leftarrow \mu_\phi(\mathcal{S}_t)$ {evaluate actor Eq. (12)}
8:          $\tilde{\mathcal{A}}_t \leftarrow \mathcal{A}_t + e_t$, $e_t \sim \mathbb{P}(E)$ {Eq. (13)}
9:          $r_t \leftarrow r(s_t, c_{\tilde{\mathcal{A}}_t})$
10:         $s_{t+1} \leftarrow$ execute action $c_{\tilde{\mathcal{A}}_t}$
11:         Store transition $(s_t, \tilde{\mathcal{A}}_t, r_t, s_{t+1})$ in $\mathcal{B}$
12:         Sample batch of $n$ transitions from $\mathcal{B}$
13:         Compute targets $y_i$ using Eq. (3)
14:         Update critic using Prop. 3.1
15:         Update actor using Def. 3.3
16:      **end for**
17: **end for**

---

$$\mathcal{Q}_t = \langle c_{\mathcal{Q}_t}, G_{\mathcal{Q}_t} \rangle_Z = \texttt{enclose}(Q_\theta, \mathcal{S}_t \times \tilde{\mathcal{A}}_t), \tag{13}$$

where the Cartesian product ($\times$) respects the dependencies between two zonotopes (Lützow & Althoff, 2023, Sec. II.C). The environment receives the perturbed center of the action set $c_{\tilde{\mathcal{A}}_t} := c_{\mathcal{A}_t} + e_t$ and returns the reward $r(s_t, c_{\tilde{\mathcal{A}}_t})$ and next state $s_{t+1}$, which are stored in the replay buffer $\mathcal{B}$ as transition $(s_t, \tilde{\mathcal{A}}_t, r_t, s_{t+1})$. To obtain temporaly uncorrelated samples, we randomly sample $n$ transitions from the buffer for the training of the critic neural network. For each transition $i \in [n]$, we compute the target $y_i$ using (3) and compute gradient sets using a set-based loss function:

**Proposition 3.1** (Set-Based Regression Loss). *Given output set $\mathcal{Q}_i = \langle c_{\mathcal{Q}_i}, G_{\mathcal{Q}_i} \rangle_Z \subset \mathbb{R}$ and target $y_i \in \mathbb{R}$, the set-based regression loss is defined as*

$$L_{Reg}(y_i, \mathcal{Q}_i) = \underbrace{1/2(c_{\mathcal{Q}_i} - y_i)^2}_{\text{standard training loss}} + \underbrace{\eta_Q/\epsilon \ \text{lnDia}(G_{\mathcal{Q}_i})}_{\text{robustness}},$$

*with weighting factor $\eta_Q \in \mathbb{R}_+$ and perturbation radius $\epsilon \in \mathbb{R}_+$. The gradient of $L_{Reg}$ w.r.t. $\mathcal{Q}_i$ is:*

$$\nabla_{\mathcal{Q}_i} L_{Reg}(y_i, \mathcal{Q}_i) = \langle c - y_i, \eta_Q/\epsilon \ \text{lnDia}'(G_{\mathcal{Q}_i}) \rangle_Z.$$

*Proof.* See Appendix B. $\square$

The set-based regression loss combines the half-squared error (Bishop & Nasrabadi, 2006, Eq. 5.14) of the center with a robustness loss to reduce the size of the output set. The actor neural network is trained using a set-based policy gradient which contains a different gradient for each possible output.

**Definition 3.2** (Set-Based Policy Gradient *SA-SC*). *Given states $\mathcal{S}_i$ with the corresponding actions $\mathcal{A}_i = \langle c_{\mathcal{A}_i}, G_{\mathcal{A}_i} \rangle_Z$ and critic outputs $\mathcal{Q}_i = \langle c_{\mathcal{Q}_i}, G_{\mathcal{Q}_i} \rangle_Z$, a set-based policy gradient is defined as*

$$\nabla_{\mathcal{A}_i} J_{Set}(\mu_\phi) := \left\langle \underbrace{\nabla_{c_{\mathcal{A}_i}} J_{Set}(\mu_\phi)}_{\text{policy gradient}}, \underbrace{\nabla_{G_{\mathcal{A}_i}} J_{Set}(\mu_\phi)}_{\text{robustness}} \right\rangle_Z,$$

*where $\nabla_{c_{\mathcal{A}_i}} J_{Set}(\mu_\phi) = \mathbb{E}_{s_i \sim \rho^\beta} [\nabla_{c_{\mathcal{A}_i}} c_{\mathcal{Q}_i}],$*

*and $\nabla_{G_{\mathcal{A}_i}} J_{Set}(\mu_\phi) = -\eta_\mu/\epsilon \ \mathbb{E}_{s_i \sim \rho^\beta} [\omega \ \text{lnDia}'(G_{\mathcal{A}_i}) + (1 - \omega) \ \nabla_{G_{\mathcal{A}_i}} \text{lnDia}'(G_{\mathcal{Q}_i})],$*

*with weights $\eta_\mu \in \mathbb{R}_+$, $\omega \in [0, 1]$ and perturbation $\epsilon \in \mathbb{R}_+$.*

The set-based policy gradient is a zonotope consisting of a center, which corresponds to the standard policy gradient (4), and a generator matrix. The generator matrix $\nabla_{G_{\mathcal{A}_i}} J_{Set}(\mu_\phi)$ uses the factor $\omega$ to connect a robustness loss for the action set $\mathcal{A}_i$ and the gradients of the robustness loss of the critic outputs $\mathcal{Q}_i$ in the action space (Fig. 2). Similar to (Zhang et al., 2021a, Sec. 3.3), the set-based policy gradient reduces the size of the action set to mitigate performance loss caused by an adversary. Finally, given the gradients w.r.t. the output of the actor

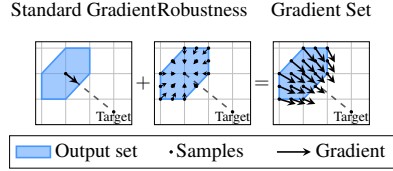

Standard Gradient  Robustness  Gradient Set

| Output set | · Samples | → Gradient |

Figure 3: Visualization of the policy gradient set (Def. 3.3): each point in the output set has a different gradient.

and the critic, we can update the respective parameters using set-based backpropagation (Koller et al., 2025, Sec. IV-B.). We can omit the set propagation through the critic with $\omega = 1$. In this case, only the robustness loss of the action set is used (*SA-PC*; set-evaluation of ② in Fig. 2). Fig. 3 visualizes the gradient set for samples of a zonotope: The gradients of the robustness loss point toward the center and thereby enforce smaller output sets.

**Definition 3.3** (Set-Based Policy Gradient *SA-PC*). *Given states $\mathcal{S}_i$ with the corresponding actions $\mathcal{A}_i = \langle c_{\mathcal{A}_i}, G_{\mathcal{A}_i} \rangle_Z$, a set-based policy gradient is defined as*

$$\nabla_{\mathcal{A}_i} J_{Set}(\mu_\phi) := \left\langle \underbrace{\nabla_{c_{\mathcal{A}_i}} J_{Set}(\mu_\phi)}_{\text{policy gradient}}, \underbrace{\nabla_{G_{\mathcal{A}_i}} J_{Set}(\mu_\phi)}_{\text{robustness}} \right\rangle_Z,$$

*where $\nabla_{c_{\mathcal{A}_i}} J_{Set}(\mu_\phi) = \mathbb{E}_{s_i \sim \rho^\beta} [\nabla_{c_{\mathcal{A}_i}} Q_\theta(s_i, c_{\mathcal{A}_i})], \nabla_{G_{\mathcal{A}_i}} J_{Set}(\mu_\phi) = -\eta_\mu/\epsilon \ \mathbb{E}_{s_i \sim \rho^\beta} [\text{lnDia}'(G_{\mathcal{A}_i})],$*

*with weight $\eta_\mu \in \mathbb{R}_+$ and perturbation $\epsilon \in \mathbb{R}_+$.*

## 4 DERIVATION OF SET-BASED LOSS FUNCTIONS

In this section, we motivate our choices for the set-based loss function (Prop. 3.1) and our set-based policy gradients (Def. 3.2 and 3.3) that are used to computed the gradient sets. While maximizing the likelihood of outputs is a standard procedure to derive loss functions (Bishop & Nasrabadi, 2006, Sec. 1.2.5), lifting this theory to set-based computing has the unique advantage of integrating formal methods into the training process. To this end, we connect set-based computing and probability theory by assuming a probability distribution over the considered closed sets. For our derivations, we use a conditional posterior distribution which can be rewritten to be proportional to a likelihood function and a prior distribution (Bishop & Nasrabadi, 2006, Eq. 1.44):

$$\text{cond. posterior} \propto \text{likelihood} \cdot \text{prior}. \tag{14}$$

This reformulation is used as the posterior and is not directly obtainable, but we can assume distributions for the likelihood and the prior to obtain an estimate. In our case, the likelihood corresponds to the standard (point-based) training goal, and the prior penalizes the volume of the computed sets. As zonotopes represent these sets and thus are point-symmetric, we additionally assume that the expected value over a zonotope is its center:

$$\mathbb{E}_{z \sim \mathcal{Z}}[z] = c. \tag{15}$$

### 4.1 SET-BASED REGRESSION LOSS

We sample random transitions $i \in [n]$ from the buffer $\mathcal{B}$ to obtain a state $s_i$ and the corresponding actions $\tilde{a}_i \sim \tilde{\mathcal{A}}_i$. For each transition, we use (13) to obtain the critic output $\mathcal{Q}_i$ and use (3) to obtain the target $y_i$ for each critic output $q_i \sim \mathcal{Q}_i$ using the rewards and next states stored in the buffer. To train $Q_\theta$, we want to maximize the probability $p_\theta(q_i|y_i, s_i, \tilde{a}_i, \beta^{-1})$. Since this probability can not be computed directly, we model this probability as a conditional posterior using (14):

$$\underbrace{p_\theta(q_i|y_i, s_i, \tilde{a}_i, \beta^{-1})}_{\text{cond. posterior}} \propto \underbrace{p(y_i|q_i, \beta^{-1})}_{\text{likelihood}} \underbrace{p_\theta(q_i|s_i, \tilde{a}_i)}_{\text{prior}}. \tag{16}$$

For the prior, we assume that $q_i$ is uniformly distributed over the interval $[l_{\mathcal{Q}_i}, u_{\mathcal{Q}_i}] \supseteq \mathcal{Q}_i \subset \mathbb{R}$, as Prop. 2.3 is also defined over the enclosing interval. Thus, the prior is given by

$$p_\theta(q_i|s_i, \tilde{a}_i) = \mathscr{U}(q_i|l_{\mathcal{Q}_i}, u_{\mathcal{Q}_i}) = \text{dia}(\mathcal{Q}_i)^{-1}. \tag{17}$$

As the critic learns the bootstrapped Q-iterations by regression, we assume that $y_i$ is normally distributed with mean $q_i$ and variance $\beta^{-1}$ with the likelihood (Bishop & Nasrabadi, 2006, Eq. 1.60):

$$p(y_i|q_i, \beta^{-1}) = \mathscr{N}(y_i|q_i, \beta^{-1}) = \sqrt{\beta/2\pi} \exp\left(-\beta/2 \left(q_i - y_i\right)^2\right). \tag{18}$$

Since we observe not a single $q_i$ but an entire set $\mathcal{Q}_i$, we use the expected value $\mathbb{E}_{q_i \sim \mathcal{Q}_i}[q_i] = c_{\mathcal{Q}_i}$ (15) in the likelihood function (Bishop & Nasrabadi, 2006, Sec. 10.3). Thus, we obtain the following term to be maximized:

$$p_\theta(q_i|y_i, s_i, \tilde{a}_i, \beta^{-1}) \propto p(y_i|c_{\mathcal{Q}_i}, \beta^{-1}) p_\theta(q_i|s_i, \tilde{a}_i) \tag{19}$$

Finally, we apply the negative logarithm and set $\beta = (\eta_{\mathcal{Q}}/\epsilon)^{-1}$ to obtain our set-based loss (Prop. 3.1):

$$-\ln\left(p(y_i|c_{\mathcal{Q}_i}, \beta^{-1}) p_\theta(q_i|s_i, \tilde{a}_i)\right) \overset{(18),(17)}{=} -\ln\left(\mathscr{N}(y_i|q_i, \beta^{-1}) \text{dia}(\mathcal{Q}_i)^{-1}\right)$$
$$\propto \beta/2 \left(c_{\mathcal{Q}_i} - y_i\right)^2 + \ln\text{Dia}(G_{\mathcal{Q}_i}) \overset{\text{Prop. 3.1}}{\propto} L_{Reg}(y_i, \mathcal{Q}_i). \tag{20}$$

We choose $\beta$ that way for easier fine-tuning (Koller et al., 2025, Def. 5).

### 4.2 SET-BASED POLICY GRADIENT

For a state $s_i$, we derive the set-based policy gradient analogous to (Xiao & Wang, 2022), to maximize the probability of an action $a_i \sim \mathcal{A}_i$ being optimal given $s_i$ – which is again not directly obtainable. Thus, we introduce a binary variable $o_i$ indicating whether $a_i$ is optimal and abbreviate $o_i = 1$ by $o_i$ (Xiao & Wang, 2022, Sec. 3.1). We again model this probability as a conditional posterior over the action output using (14):

$$\underbrace{p_{\phi,\theta}(a_i|o_i, s_i, q_i, \alpha)}_{\text{cond. posterior}} \propto \underbrace{p(o_i|q_i, \alpha)}_{\text{likelihood}} \underbrace{p_{\phi,\theta}(a_i|q_i, s_i)}_{\text{prior}}. \tag{21}$$

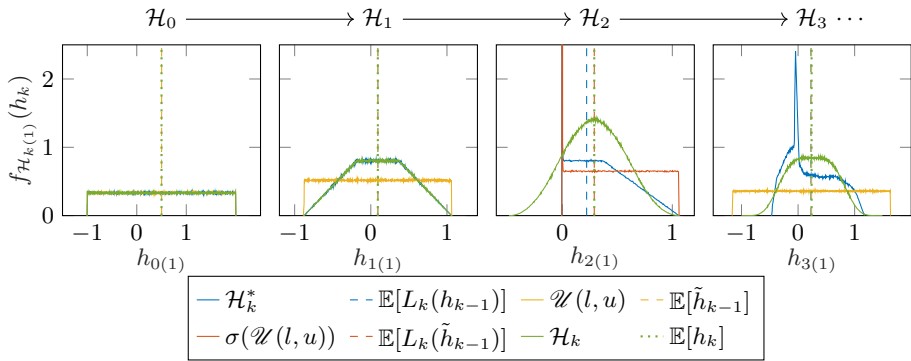

Figure 4: Propagated probability density func. $f_{\mathcal{H}_{k(1)}}(h_k)$ of zonotopes for a ReLU-neural net: True sampled density func.(blue), interval enclosure (yellow), and density of sets from Prop. 2.3 with $\beta_j \sim \mathscr{U}(-1, 1)$ (Def. 2.2) (green).

We assume the likelihood to be exponentially distributed with $\alpha \in \mathbb{R}_+$ (Xiao & Wang, 2022, Sec. 3.2):

$$p(o_i|a_i, q_i, \alpha) = \exp(\alpha^{-1} q_i). \tag{22}$$

The prior in (21) is not directly computable. Thus, we model it again as a conditional posterior (14):

$$\underbrace{p_{\phi,\theta}(a_i|q_i, s_i)}_{\text{cond. posterior}} = \underbrace{p_\theta(q_i|a_i, s_i)}_{\text{likelihood}} \underbrace{p_\phi(a_i|s_i)}_{\text{prior}}, \tag{23}$$

where the likelihood function of $q_i$ and the prior for $a_i$ are uniform distributions over the enclosing intervals $[l_{\mathcal{Q}_i}, u_{\mathcal{Q}_i}] \supseteq \mathcal{Q}_i \subset \mathbb{R}$ and $[l_{\mathcal{A}_i}, u_{\mathcal{A}_i}] \supseteq \mathcal{A}_i \subset \mathbb{R}^{n_{\mathcal{A}_i}}$ analogous to (17):

$$p_\theta(q_i|a_i, s_i) = \mathscr{U}(q_i|l_{\mathcal{Q}_i}, u_{\mathcal{Q}_i}) = \text{dia}(\mathcal{Q}_i)^{-1},$$
$$p_\phi(a_i|s_i) = \mathscr{U}(a_i|l_{\mathcal{A}_i}, u_{\mathcal{A}_i}) = \prod_{j=1}^{n_{\mathcal{A}_i}} \text{dia}(\mathcal{A}_i)_{(j)}^{-1}. \tag{24}$$

Please note that these two probabilities correspond to the evaluation of the actor and the critic, respectively. For the likelihood function of (21), the expected value $\mathbb{E}_{q_i \sim \mathcal{Q}_i}[q_i] = c_{\mathcal{Q}_i}$ (15) is used, and taking the logarithm obtains us:

$$\ln(p(o_i|c_{\mathcal{Q}_i}, \alpha)\, p_\phi(a_i|s_i)\, p_\theta(q_i|a_i, s_i)) \stackrel{(22),(24)}{=} \alpha^{-1} c_{\mathcal{Q}_i} - \mathbf{1}^\top \ln\text{Dia}(G_{\mathcal{A}_i}) - \ln\text{Dia}(G_{\mathcal{Q}_i}). \tag{25}$$

The set-based policy gradient for SA-SC (Def. 3.2) is derived by differentiation, where we again set the weighting factor $\alpha = (\eta_\mu/\epsilon)^{-1}$ for easier fine-tuning (Koller et al., 2025, Def. 5). Moreover, we introduce a factor $\omega \in [0, 1]$ to weigh the gradients of the prior terms of $\mathcal{A}_i$ and $\mathcal{Q}_i$.

**Derivation of SA-PC.** For *SA-PC*, only the actor is trained using set-based training while the critic uses standard (point-based) training (① in Fig. 2). Thus, we drop the prior for the critic output $q_i$ as this is no longer evaluated set-based and use the expected value $\mathbb{E}_{a_i \sim \mathcal{A}_i}[a_i] = c_{\mathcal{A}_i}$ for the likelihood:

$$\underbrace{p(a_i|o_i, s_i, \alpha, \phi)}_{\text{cond. posterior}} \propto \underbrace{p(o_i|s_i, c_{\mathcal{A}_i}, \alpha)}_{\text{likelihood}} \underbrace{p_\phi(a_i|s_i)}_{\text{prior}}. \tag{26}$$

Taking the logarithm while keeping our assumption on the likelihood function and the prior leads to

$$\ln(p(o_i|s_i, c_{\mathcal{A}_i}, \alpha)\, p_\phi(a_i|s_i)) \stackrel{(22),(24)}{=} \alpha^{-1} Q_\theta(s_i, c_{\mathcal{A}_i}) - \mathbf{1}^\top \ln\text{Dia}(G_{\mathcal{A}_i}). \tag{27}$$

The set-based policy gradient for SA-PC (Def. 3.3) is derived by differentiation and choosing $\alpha$ as above. This also corresponds to setting $\omega = 1$ in Def. 3.2.

### 4.3 EXPECTATION PRESERVING IMAGE ENCLOSURE

For (19), (25) and (27), we simplify the likelihood with the expected value of the neural network output, i.e., the center. This simplification is justified with Prop. 4.1. In Fig. 4, we plot the probability

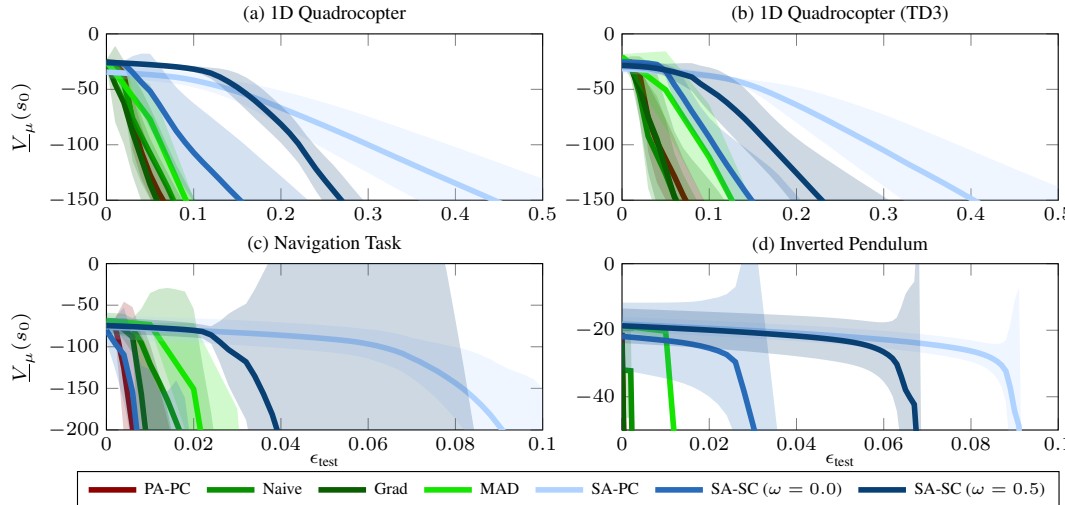

Figure 5: Comparison of verified performance $\underline{V}_\mu(s_0)$ for the (a) *1D Quadrotor*, (c) *Navigation Task*, and (d) *Inverted Pendulum*. The TD3 implementation is compared in (b) for the *1D Quadrotor*.

distributions of a set propagation and visualize the expected value for the first neuron of each layer. We compare the empirically evaluated density of a uniform input disturbance (blue) with the corresponding set-based zonotope propagation (green) across the network layers. The set propagation can be interpreted as a form of variational inference in which the output is constrained to be a zonotope. To illustrate the mean-preserving property of this single-layer variational approximation, we additionally plot the interval enclosures before activation (yellow) and the transformed intervals after activation (red). The expected value is trivially preserved for linear layers as the affine transformation for zonotopes is computed in closed form. For nonlinear layers, we observe that the expected value of the interval enclosure (red vertical line in third plot) is preserved through the enclosure (green vertical line; Prop. 2.3), with only small deviations to the expected value of the empirically evaluated density function (blue). Let us formally state this observation:

**Proposition 4.1** (Tight Expectation-Preserving Set Propagation). *Given a neural network with* ReLU-*activations and an input set* $\mathcal{H}_{k-1}$ *with the enclosing interval* $[l_{k-1}, u_{k-1}] \supseteq \mathcal{H}_{k-1}$, *the expected value of the enclosure of the* $k$-*th layer is*

$$\mathbb{E}_{h_k \sim \mathcal{H}_k}[h_k] = \mathbb{E}_{h_{k-1} \sim \mathcal{U}(l_{k-1}, u_{k-1})}[L_k(h_{k-1})],$$

*with* $\mathcal{H}_k = \texttt{enclose}(L_k, \mathcal{H}_{k-1})$ *having minimal approximation errors.*

*Proof.* See Appendix B. □

## 5 EVALUATION

We use the MATLAB toolbox CORA (Althoff, 2015) to implement our novel set-based reinforcement learning algorithm and compare the *SA-PC* and the *SA-SC* implementation against standard (point-based) training (*PA-PC*) and three state-of-the-art adversarial methods: *Naive-*, *Grad-* and *MAD* (Maximum Action Difference)-based implementations (Pattanaik et al., 2018, Alg. 2 and 4)(Zhang et al., 2020), which compute adversarial attacks to approximate the worst-case observation within a perturbation set. Similar to previous work (Yuan et al., 2022; Krasowski et al., 2023) on robust reinforcement learning, we consider four benchmarks in our evaluation: *Navigation Task*, *1D Quadrotor*, *Inverted Pendulum*, and *2D Quadrotor*. A detailed description of all benchmarks can be found in Appendix A. As in previous work (Yuan et al., 2022; Krasowski et al., 2023), we use neural networks with ReLU activations and two hidden layers of 64 and 32 neurons for the actor and critic networks. The output layer of the actor has a tanh activation. We provide the mean results and the 95% confidence interval across five different random seeds. Hyperparameters and evaluation details are given in Appendix A.

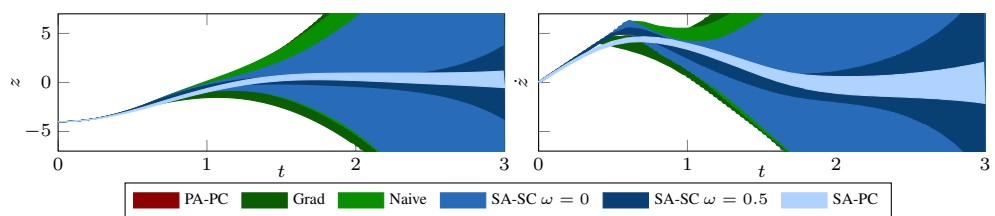

Figure 6: *Quadcopter 1D*: Comparison of reachable altitudes and vertical speeds for $\epsilon_{\text{test}} = 0.15$.

**Verified performance.** Following our problem statement (Sec. 2.6), we evaluate our agents based on their worst-case cumulative reward given uncertainties, which we compute using reachability analysis in CORA. Starting at an initial point $s_0$, reachability analysis encloses all reachable states within a time interval $[0, t_{\text{end}}]$ with time horizon $t_{\text{end}} \in \mathbb{R}_+$ (Fig. 1b). Uncertainties (11) are added at each time step $t \in \{0, \ldots, t_{\text{end}}\}$ and carried through until the time horizon is reached. Please note that, in general, this process is outer-approximative due to the continuous time and nonlinearities within the system and networks. Thus, reachability analysis obtains a formally verified lower bound of the worst-case cumulative reward, which we refer to as *verified performance* from now on. For reward functions of the form $r(s_t, a_t) = w^\top |s_t - s^*|$, we can use set-based computing to obtain the verified performance $\underline{V}_\mu(s_0)$ computed from the set of states $\mathcal{S}_t = \langle c_t, G_t \rangle_Z$ obtained by CORA:

$$\underline{V}_\mu(s_0) = \sum_{t=0}^{t_{\text{end}}} \gamma^t \max_{s_t \in \mathcal{S}_t} w^\top |s_t - s^*| \overset{(5),(6)}{=} \sum_{t=0}^{t_{\text{end}}} \gamma^t \big(w^\top |c_t - s^*| + \text{dia}(w^\top \mathcal{S}_t)/2\big). \quad (28)$$

Intuitively, $\underline{V}_\mu(s_0)$ is high if we can formally verify the robustness of our agent and drops otherwise. Previous works evaluated their agents solely through adversarial attacks, with learned probabilistic dynamic models (Yang et al., 2024) or randomized smoothing (Wu et al., 2022), which provide an empirical upper bound or a probabilistic bound of the worst-case reward and thus do not determine a formal lower bound.

**Main results.** We present the verified performance with increasing perturbation radius $\epsilon_{\text{test}}$ (11) of the considered training methods in Fig. 5. The set-based algorithms *SA-PC* and *SA-SC* train agents that can be verified for up to 9 times larger perturbation radii $\epsilon_{\text{test}}$ than the other methods. Thus, we can show robust performance of those agents, although much larger disturbances are considered. We showcase this in Fig. 6 using a large perturbation radius, where the reachable set of the *SA-PC* agent remains much smaller and thus the stability can be formally verified. As $\epsilon_{\text{test}}$ increases, the set-based training methods show a higher verified performance across all benchmarks, indicating reduced performance degradation under growing disturbance (Fig. 5).

**Ablation study.** Let us continue with our ablation study on various components of our algorithm:

*1) Influence of weighting factor $\omega$ (Def. 3.2):* Recall that this parameter is used to determine where the volume of the set is penalized: With $\omega = 0$, only the output set of the critic is penalized (*SA-SC*) and with $\omega = 1$, only the output set of the actor is penalized, which corresponds to the *SA-PC* method. The *SA-PC* implementation trains more conservative actors, which perform best for large $\epsilon_{\text{test}}$ while having a worse verified performance for small $\epsilon_{\text{test}}$. For example, the *SA-PC* actor for the *1D Quadrotor* is the most robust but reaches the goal later using lower vertical velocities (Fig. 6). By fine-tuning $\omega$, a balance can be found where the verified performance for small $\epsilon_{\text{test}}$ can be regained while still being much more robust for larger $\epsilon_{\text{test}}$ (Fig. 5).

*2) Performance under different attacks:* It is well-known that networks trained using adversarial attacks are robust against their respective attack method but might not be robust against other methods. We observe in Fig. 7 that under *Naive* and *Grad* attacks, also their respectively trained agents outperform standard point-based training with DDPG Lillicrap et al. (2016). In particular, *SA-SC* with $\omega = 0$ and $\omega = 0.5$, perform on-par with these attacks, while *SA-PC* trained actors are more conservative and under-perform under these attacks. This additionally motivates the modularity of *SA-SC*, which is investigated in ablation *1)*. However, verified performance captures all attacks and, thus, *Grad*, *Naive*, and *MAD* methods perform similarly to standard point-based training (*PA-PC*) across all benchmarks (Fig. 5), while *SA-SC* and *SA-PC* demonstrate their robustness against the entire set of all possible perturbations.

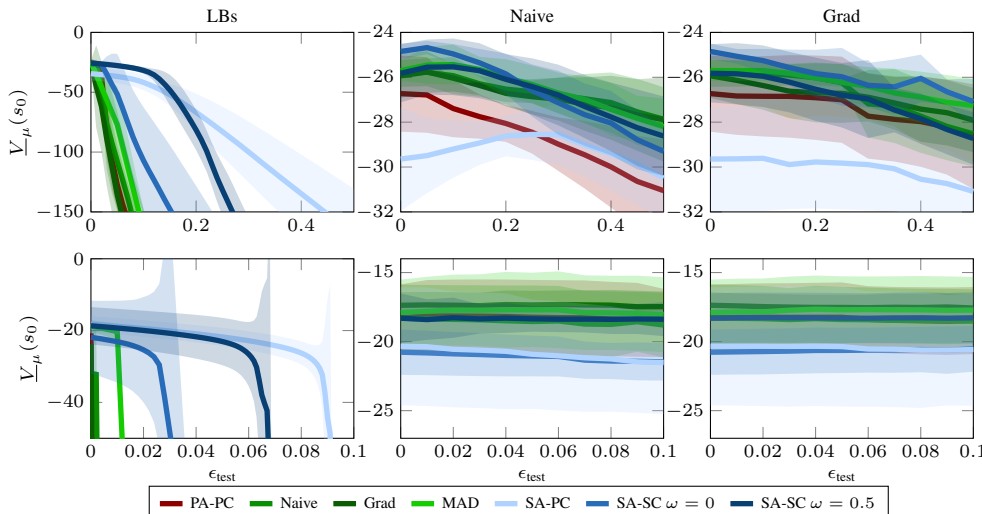

Figure 7: (Top) *Quadcopter 1D* and (bottom) *Inverted Pendulum*: Comparison of the return under *Naive* and *Grad* attacks with $\underline{V}_\mu(s_0)$.

Table 1: Time $[s]$ for verification of Navigation task agents. Entries with $-$ indicate that the verification toolbox is not able to verify the system.

| Toolbox | CORA | CROWN-Reach | JuliaReach | NVV |
|---|---|---|---|---|
| Standard | 423.45 | 130.12 | — | — |
| Set-Based | 1.99 | 20.65 | 4.49 | 1903.24 |

*3) Performance under different verification methods:* Since different formal verification techniques may already be in place for various safety-critical systems, we also show in Tab. 1 that our set-based trained agents can be verified using alternative methods that do not rely on zonotopes. All toolboxes (Althoff, 2015; Xiangru Zhong, 2024; Bogomolov et al., 2019; Tran et al., 2020) verified the set-based agent, while only two verified the standard trained agent, taking much longer to do so. Hence, our set-based agents are easier to verify.

*4) Extension to ensemble methods:* Our proposed set-based reinforcement learning can be directly extended for the Twin Delayed Deep Deterministic policy gradient algorithm (TD3) (Fujimoto et al., 2018) and other ensemble algorithms (Januszewski et al., 2021). Fig. 5b shows overall a similar performance as with DDPG and for the *SA-PC* version, a better performance for small perturbation radii $\epsilon_{test}$ of the set-based TD3 algorithm for the *1D Quadrotor*.

*5) Scalability:* In Appendix A.3, we show that our algorithm scales to more complex benchmarks with intricate dynamics and larger state and action spaces (Todorov et al., 2012). While the worst-case reward cannot be directly evaluated for every benchmark, we show that for systems lacking formal verification toolboxes, our algorithm scales effectively in practice and obtains robust neural networks.

## 6 CONCLUSION

We introduce the first set-based reinforcement learning algorithm. Unlike other algorithms that rely on adversarial inputs, our approach is unique in using set-based neural network training, working with entire sets of inputs and gradients. The gradient sets used for training are computed from set-based loss functions that are motivated by a rigorous analysis of the underlying set propagation. Our experimental results on different benchmarks demonstrate the efficacy of our approach. Particularly, our trained controllers can be formally verified for large perturbation sets, which is essential for their deployment in safety-critical environments. Consequently, set-based reinforcement learning is an effective, novel approach for training robust neural network controllers.

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

## A  EVALUATION DETAILS

### A.1  BENCHMARK DESCRIPTIONS

Let us briefly describe the specifications of the remaining benchmarks in this section.

*1D Quadrotor:* The state is $s = \begin{bmatrix} z & \dot{z} \end{bmatrix}^\top$, with altitude $z$, velocity $\dot{z}$, and dynamics (Yuan et al., 2022):

$$\dot{s} = \begin{bmatrix} \dot{z} & \frac{a+1}{2\,m} - g \end{bmatrix}^\top, \tag{29}$$

with action space $a \in [-1, 1]$, gravity $g = 9.81$, and mass $m = 0.05$. Starting from initial states $s_0 \in [[-4 \quad 0]^\top, [4 \quad 0]^\top]$, the Quadrotor is stabilized at $s^* = \mathbf{0}$; the reward function is $r(s_t, a_t) = -[1 \quad 0.01]|s_{t+1} - s^*|$ for a time horizon of $3s$.

*Navigation Task:* We use a unicycle model with states $s = \begin{bmatrix} x & y & \theta & v \end{bmatrix}^\top$ and dynamics (Lopez et al., 2022):

$$\dot{s} = \begin{bmatrix} v \cos(\theta) & v \sin(\theta) & a_{(1)} & a_{(2)} \end{bmatrix}^\top, \tag{30}$$

with action space $a \in [-1, 1]$. From starting point $s_0 = \begin{bmatrix} 3 & 3 & 0 & 0 \end{bmatrix}^\top$, the task is to navigate to the goal $s^* = \mathbf{0}$ without colliding with an obstacle $\mathcal{O} = [\mathbf{1}, 2 \cdot \mathbf{1}]$. The reward function is

Table 2: Training parameters for *PA-PC*, *SA-PC* and *SA-SC*.

| Parameter | DDPG | TD3 |
|---|---|---|
| Actor learning rate | $1 \cdot 10^{-4}$ | $1 \cdot 10^{-4}$ |
| Critic learning rate | $1 \cdot 10^{-3}$ | $1 \cdot 10^{-3}$ |
| Critic $L_2$ weight regularization $\lambda_Q$ | 0.01 | 0 |
| Discount factor $\gamma$ | 0.99 | 0.99 |
| Target update factor $\tau$ | 0.05 | 0.05 |
| Exploration noise std. deviation $\sigma$ | 0.1 | 0.1, 0.2 |
| Batchsize | 64 | 64 |
| Buffersize | $1 \cdot 10^6$ | $1 \cdot 10^6$ |
| Episodes | 2000 | 2000 |
| Perturbation radius $\epsilon$ | 0.1 | 0.1 |
| Actor weighting factor $\eta_\mu$ | 0.1 | 0.1 |
| Critic weighting factor $\eta_Q$ | 0.01 | 0.01 |

$r(s_t, a_t) = -[1 \quad 1 \quad 0 \quad 0]\,|s_{t+1} - s^*| - c$, with $c = 1$ if $s_{t+1} \in \mathcal{O}$ and otherwise $c = 0$. We consider a time horizon of $8s$.

*Inverted Pendulum:* The state is $s = \begin{bmatrix} \theta & \dot\theta \end{bmatrix}^\top$, with angle $\theta$, angular velocity $\dot\theta$, dynamics (Krasowski et al., 2023):

$$\dot s = \begin{bmatrix} \dot\theta & \frac{g}{l}\sin(\theta) + \frac{1}{m\,l^2}\,a \end{bmatrix}^\top, \tag{31}$$

with action space $a \in [-15, 15]$, gravity $g = 9.81$, mass $m = 1$, and length $l = 1$. The goal to stabilize the pendulum in the upright position $s^* = \mathbf{0}$; the reward function is $r(s_t, a_t) = -[1 \quad 0.01]\,|(s_{t+1} - s^*)|$ for a time horizon of $3s$.

*2D Quadrotor:* The state of the system is defined as $s = \begin{bmatrix} x & \dot x & z & \dot z & \theta & \dot\theta \end{bmatrix}^\top$, with horizontal displacement $x$, horizontal velocity $\dot x$, altitude $z$, vertical velocity $\dot z$, angle $\theta$, angular velocity $\dot\theta$ and dynamics (Yuan et al., 2022):

$$\dot s = \begin{bmatrix} \dot x \\ \sin(\theta)\frac{\tilde a_{(1)} + \tilde a_{(2)}}{m} \\ \dot z \\ \cos(\theta)\frac{\tilde a_{(1)} + \tilde a_{(2)}}{m} - g \\ \dot\theta \\ \frac{l(\tilde a_{(2)} - \tilde a_{(1)})}{\sqrt{2}J_y} \end{bmatrix}, \tag{32}$$

where $\tilde a = (1 + \frac{1}{2} a)\frac{m\,g}{2}$ with action $a \in [-\mathbf{1}, \mathbf{1}] \subset \mathbb{R}^2$. The constant $g = 9.81$ defines gravity, $m = 0.027$ is the mass of the Quadrotor, $l = 0.0397$ and $J_y = 1.4 \cdot 10^{-4}$ defines the arm length of the propeller mount and the moment of inertia around the $y$ axis. The reward function is given by $r(s_t, a_t) = -[1 \quad 0.01 \quad 1 \quad 0.01 \quad 0 \quad 0]|s_{t+1} - s^*|$ for a time horizon of $3s$, with the goal to stabilize the Quadrotor at $s^* = \mathbf{0}$.

## A.2 Hyperparameters

We give the hyperparameters used to train the networks in our evaluation in Tab. 2.

## A.3 Additional Experiments

**Quadrotor 2D:** We provide additional experiments on the *2D Quadrotor* and the MuJoCo *Hopper-v2* benchmark (Todorov et al., 2012). We again average the results over the last five agents in each training run and compute the mean and a $95\%$ confidence interval across five independent random seeds. Fig. 8 compares the lower bounds $\underline{V}_\mu(s_0)$ of the different training algorithms.

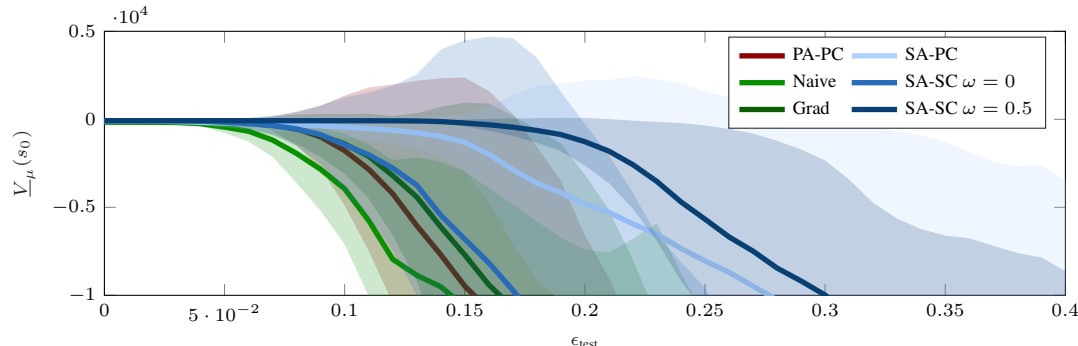

Figure 8: Comparison of $\underline{V}_\mu(s_0)$ for Quadrotor 2D.

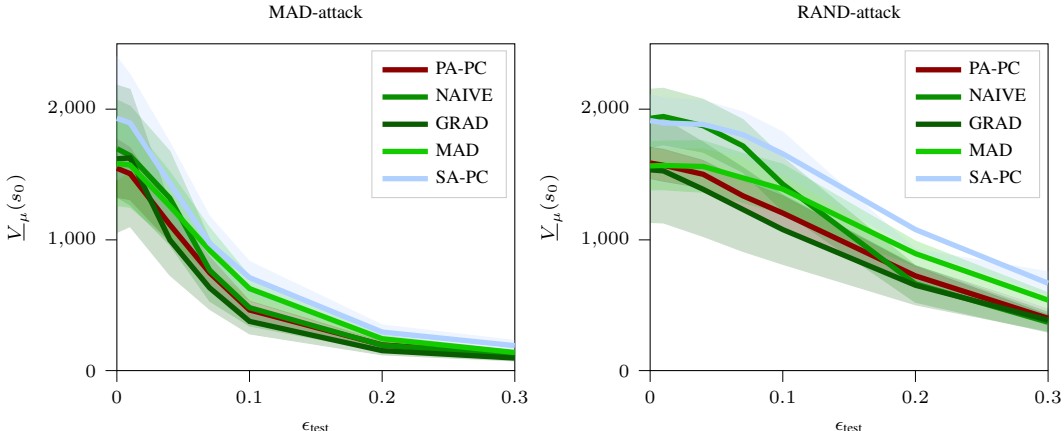

Figure 9: Comparison of $\underline{V}_\mu(s_0)$ for *Hopper-v2* approximated by MAD (Zhang et al., 2020).

Figure 10: Comparison of $\underline{V}_\mu(s_0)$ for *Hopper-v2* approximated with uniform random noise.

**Locomotion:** Since, to our best knowledge, no existing verification toolbox can compute reachable sets for locomotion benchmarks, we are unable to provide a formal lower bound for *Hopper-v2*. However, we want to stress this is not a limitation of our training approach but rather of the subsequent verification step. Thus, to demonstrate the scalability of our approach to such benchmarks, we approximate $\underline{V}_\mu(s_0)$ using 50 trajectories under the MAD-attack (Zhang et al., 2020) and 200 trajectories perturbed with uniform random noise sampled from the $\ell_\infty$ ball with radius $\epsilon_{\text{test}}$. The corresponding worst-case returns are shown in Figure 9 (MAD attack) and Figure 10 (random noise). For *Hopper-v2*, these empirical estimates provide an upper bound on the worst-case performance and demonstrate the scalability of our method. Notably, even under noise-free conditions, the worst returns of SA-PC trained agents, evaluated over randomly initialized trajectories, consistently exceed those of agents trained with a point-wise robustness criterion. We additionally provide videos illustrating agent behaviors under the uniform-random[2] and MAD[3] attacks using the first random seed, with $\epsilon_{\text{test}} = 0.1$ and $\epsilon_{\text{test}} = 0.075$. To better visualize the agents' failure modes, we disable early termination in these demonstrations. The results clearly show that the MAD attack is more effective at degrading agent performance compared to the uniform-random baseline. Notably, across both attack types, our *SA-PC* agent consistently demonstrates superior robustness and overall performance.

**Hyperparameter Discussion:** Set-based reinforcement learning introduces the additional parameters $\eta_\mu$ and $\eta_Q$, which appear in the set-based policy gradient Def. 3.2 and the set-based regression loss Prop. 3.1. In Tab. 2, we list the hyperparameters used in our benchmarks.. Notably $\eta_\mu$ and $\eta_Q$ are fixed in all experiments. We ease tuning these parameters by choosing $\beta$ and $\alpha$ in Eq. (20) and (25) as proposed by Koller et al. (2025). For a given perturbation radius $\epsilon_{\text{test}}$, we study for *SA-PC* on the 1D

---

[2]Video: `https://t1p.de/ar23e`
[3]Video: `https://t1p.de/mye6l`

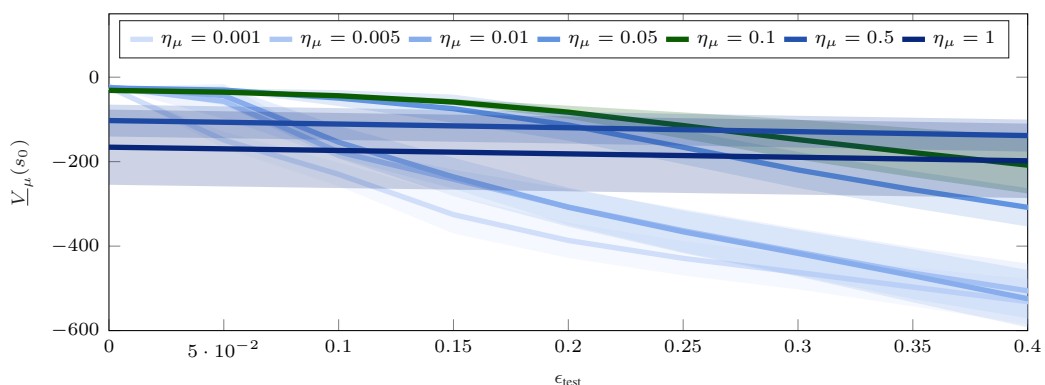

Figure 11: Tradeoff in verified performance for hyper parameter ablation $\eta_\mu$.

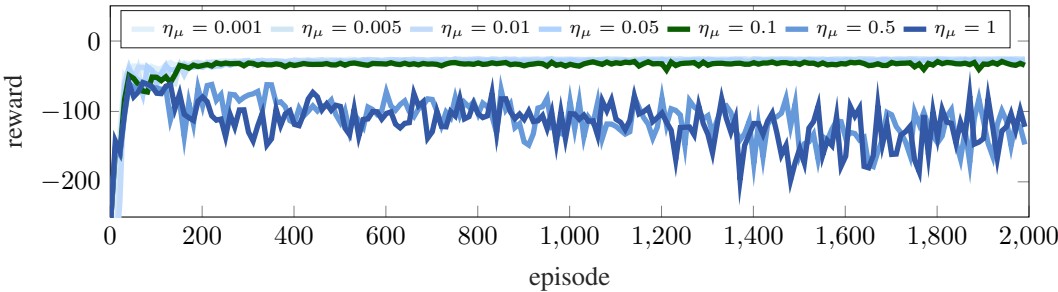

Figure 12: Learning behavior for hyper parameter ablation $\eta_\mu$.

Quadrotor benchmark, the effect of varying $\eta_\mu$. The hyperparameter $\eta_\mu$ directly scales the set-based gradient of the zonotope generators. As shown in Fig. 11, increasing $\eta_\mu$, improves robustness: verified performance near the noise-free case $\epsilon_{\text{test}} = 0$ decreases slightly for larger $\eta_\mu$, whereas verified performance at larger $\epsilon_{\text{test}}$ increases. This matches the intended trade-off between nominal (zero-perturbation) performance and robustness to larger perturbations. We therefore conclude that the choice of $\eta_\mu = 0.1$ (green) is considerably good, since it still achieves high verified rewards for large $\epsilon_{\text{test}}$, while it preserves good rewards for the noise-free case $\epsilon_{\text{test}} = 0$. We also report the learning history for different hyperparameter settings in Fig. 12. Moderate values of $\eta_\mu$ lead to stable convergence, while very large $\eta_\mu$ slow or prevent convergence to the optimum.

We next conduct an ablation study on the hyperparameter $\eta_Q$ for *SA-SC*, keeping $\eta_\mu = 0.1$ fixed. As shown in Fig. 13, large values of $\eta_Q$ (e.g. $\eta_Q = 1$)make the value function highly contractive, which leads to reduced performance for small perturbation radii $\epsilon_{\text{test}}$and to convergence difficulties, as illustrated in Fig. 14. As $\eta_Q$decreases, convergence improves and the value function captures more of the variability induced by observation noise. Consequently, the actor learns a more conservative policy, resulting in higher verified performance for larger $\epsilon_{\text{test}}$. We find that $\eta_Q = 0.01$ (green) provides the best verified performance across all perturbation radii. Further reducing $\eta_Q$, causes verified performance to decline again.

**Learning History:** In Fig. 16, we present the full learning curves, which show that set-based reinforcement learning exhibits convergence behavior comparable to the adversarial baselines. For clarity, we report the rewards evaluated without observation noise during training. Additionally, we report the learning curves for the Locomotion benchmark (Hopper-v2) from Fig. 9 and 10 in Fig. 15. For this analysis, we evaluate both algorithms using the reward without observation noise. We find that, also in the

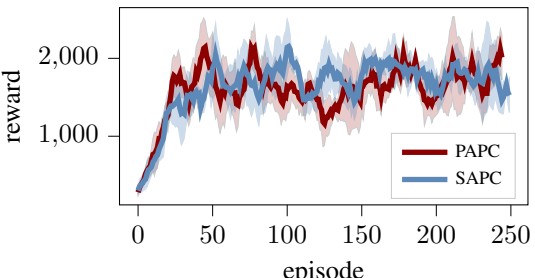

Figure 15: Full learning history for Hopper-v2.

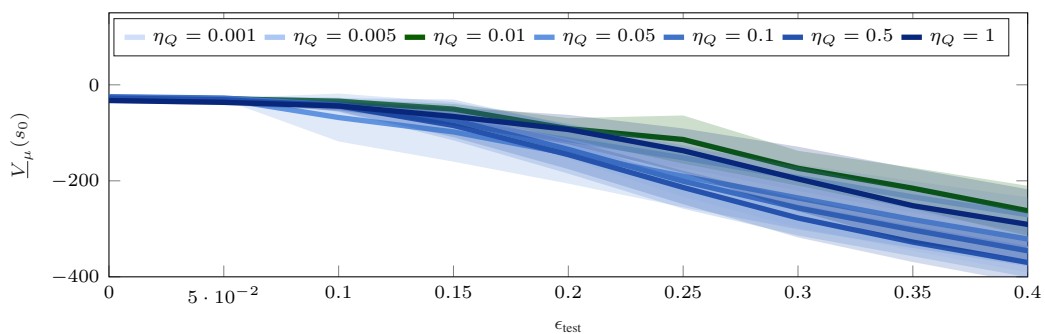

Figure 13: Hyper parameter ablation $\eta_Q$.

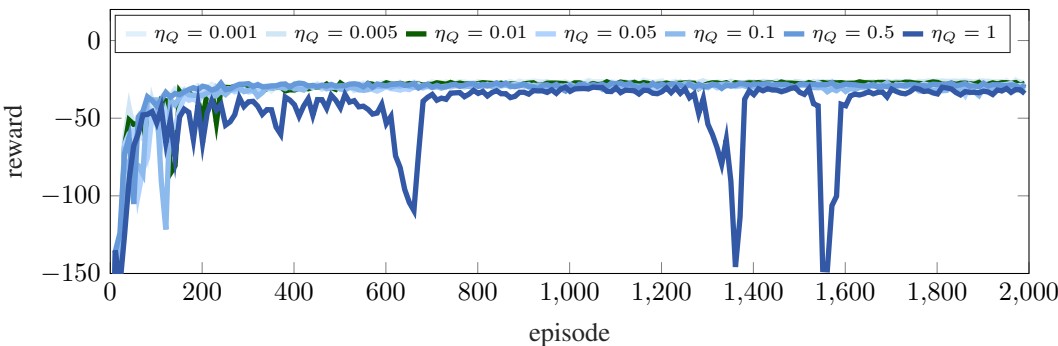

Figure 14: Learning behavior for hyper parameter ablation $\eta_Q$.

Locomotion setting, our set-based method exhibits learning dynamics similar to those of the vanilla DDPG Lillicrap et al. (2016) implementation. Based on the hyperparameter analysis in Fig. 11 and 12, we also select $\eta_\mu = 0.1$ for the *Hopper-v2* benchmark.

## A.4 RUNTIME COMPLEXITY

Building on set-based neural network training (Koller et al., 2025), our algorithm has polynomial time complexity. Based on the analysis in Koller et al. (2025, Prop. 17), we observe that *SA-PC* has runtime complexity $\mathcal{O}(n_{max,\mu}^2 q_\mu \kappa_\mu)$, where $n_{max,\mu}$ is the maximum number of neurons per layer for the actor. The sum $q = n_0 + \sum_{k \in \kappa_\mu} n_k$ includes the number of initial independent noise generators $n_0$ and the total number of neurons in the network. For *SA-SC*, we propagate the zonotopes through both networks, actor and critic. Therefore, the time complexity is hence given by

$$\mathcal{O}(n_{max,\mu+Q}^2 q_{\mu+Q} \kappa_{\mu+Q}), \tag{33}$$

where $n_{max,\mu+Q}$ is the maximum number of neurons per layer in either of actor or critic. Second, the total number of layers $\kappa_{\mu+Q} = \kappa_\mu + \kappa_Q$ is the sum of layers in the actor $\kappa_\mu$ and the critic $\kappa_Q$. Finally, $q_{\mu+Q} = n_0 + \sum_{k \in \kappa_\mu} n_k + \sum_{k \in \kappa_Q} n_k$ is now defined as the sum of the initial independent noise generators and the total number of neurons in both networks. We additionally remark that consequently our algorithm depends on $n_0$ in $q$, and thereby grows with the state dimension, if the noise is considered to be independent per dimension.

Set-based reinforcement learning can be efficiently computed batch-wise on a GPU, but the memory load remains challenging. Especially for *SA-SC*, storing entire action sets with many generators in $\mathcal{B}$ is memory-consuming and computationally more expensive. The different runtimes for 10 learning epochs are listed in Tab. 3 and were run on a server with two *AMD EPYC 7763* 64 core processors, 2 TB RAM, and an *NVIDIA A100-PCIE* 40 GB GPU.

## A.5 LIMITATIONS

We do not encode safety as an explicit constraint during training of the *Navigation Task* benchmark; instead, we incorporate safety considerations indirectly by subtracting a penalty from the reward

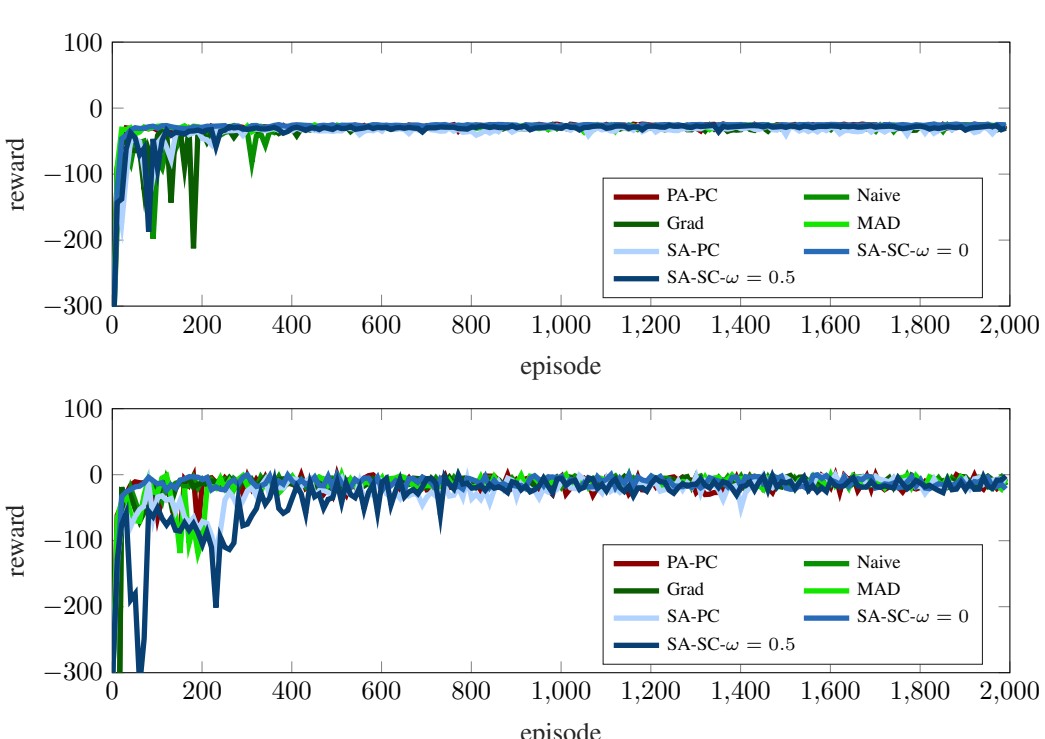

Figure 16: Full learning history for (top) 1D Quadrotor and (bottom) Inverted pendulum.

Table 3: Training times [s/10 epochs]

| Benchmark | PA-PC | Naive | Grad | SA-PC | SA-SC |
|-----------|-------|-------|------|-------|-------|
| 1D Quad. | 1.58 | 2.10 | 1.98 | 2.77 | 7.92 |
| Pendulum | 1.82 | 2.04 | 2.02 | 2.87 | 7.66 |
| Nav. Task | 2.35 | 2.89 | 2.84 | 4.35 | 12.43 |

signal. Consequently, any claim of safe behavior at deployment cannot rest solely on the provable lower bound of the cumulative reward, this bound guarantees only worst-case performance, not safety itself. To establish formal safety guarantees, the trained policy must therefore undergo separate, formal verification rather than relying on its reward lower bound.

# B  PROOFS

**Proposition 3.1.** *Given output set $\mathcal{Q}_i = \langle c_{\mathcal{Q}_i}, G_{\mathcal{Q}_i} \rangle_Z \subset \mathbb{R}$ and target $y_i \in \mathbb{R}$, the set-based regression loss is defined as*

$$L_{Reg}(y_i, \mathcal{Q}_i) = \underbrace{1/2(c_{\mathcal{Q}_i} - y_i)^2}_{\text{standard training loss}} + \underbrace{\eta_{\mathcal{Q}}/\epsilon \, \ln\text{Dia}(G_{\mathcal{Q}_i})}_{\text{robustness}},$$

*with weighting factor $\eta_Q \in \mathbb{R}_+$ and perturbation radius $\epsilon \in \mathbb{R}_+$. The gradient of $L_{Reg}$ w.r.t. $\mathcal{Q}_i$ is:*

$$\nabla_{\mathcal{Q}_i} L_{Reg}(y_i, \mathcal{Q}_i) = \langle c - y_i, \eta_Q/\epsilon \, \ln\text{Dia}'(G_{\mathcal{Q}_i}) \rangle_Z.$$

*Proof.* The gradient w.r.t. a zonotope is represented as a zonotope as well, consisting of the gradient w.r.t. the center and the gradient w.r.t. the generator matrix (Koller et al., 2025, Def. 8). Hence, the gradient of the set-based regression is:

$$
\begin{aligned}
\nabla_{\mathcal{Q}_i} L_{Reg}(y_i, \mathcal{Q}_i) \quad &= \left\langle \nabla_{c_{\mathcal{Q}_i}} L_{Reg}(y_i, \mathcal{Q}_i), \nabla_{G_{\mathcal{Q}_i}} L_{Reg}(y_i, \mathcal{Q}_i) \right\rangle_Z \\
&= \left\langle c_{\mathcal{Q}_i} - y_i, \frac{\eta_Q}{\epsilon} \text{diag}(|G_{\mathcal{Q}_i}| \mathbf{1})^{-1} \text{sign} \, G_{\mathcal{Q}_i} \right\rangle_Z \\
&= \left\langle c_{\mathcal{Q}_i} - y_i, \frac{\eta_Q}{\epsilon} \ln\text{Dia}'(G_{\mathcal{Q}_i}) \right\rangle_Z . \qquad \square
\end{aligned}
$$

**Proposition 4.1.** *Given a neural network with* ReLU-*activations and an input set $\mathcal{H}_{k-1}$ with the enclosing interval $[l_{k-1}, u_{k-1}] \supseteq \mathcal{H}_{k-1}$, the expected value of the enclosure of the $k$-th layer is*

$$\mathbb{E}_{h_k \sim \mathcal{H}_k}[h_k] = \mathbb{E}_{h_{k-1} \sim \mathcal{U}(l_{k-1}, u_{k-1})}[L_k(h_{k-1})],$$

*with $\mathcal{H}_k = \texttt{enclose}(L_k, \mathcal{H}_{k-1})$ having minimal approximation errors.*

*Proof.* We split cases on the type of layer $L_k$.

*Case (1): Linear layer* The expected value is preserved by linearity of the expectation:

$$
\begin{aligned}
\mathbb{E}_{h_k \sim \mathcal{H}_k}[h_k] &\overset{(15)}{=} c_k = W_k \, c_{k-1} + b_k = W_k \, \mathbb{E}_{h_{k-1} \sim \mathcal{U}(l_{k-1}, u_{k-1})}[h_{k-1}] + b_k \\
&= \mathbb{E}_{h_{k-1} \sim \mathcal{U}(l_{k-1}, u_{k-1})}[W_k \, h_{k-1} + b_k] = \mathbb{E}_{h_{k-1} \sim \mathcal{U}(l_{k-1}, u_{k-1})}[L_k(h_{k-1})].
\end{aligned}
$$

*Case (2): ReLU layer* Activation functions are applied element-wise, thus we consider each dimension individually; to avoid clutter, we drop the dimension index $x_{(i)}$. We distinguish between three cases: (2a) $l_{k-1}, u_{k-1} \leq 0$, (2b) $l_{k-1}, u_{k-1} \geq 0$, (2c) $l_{k-1} < 0 < u_{k-1}$. For cases (i) and (ii), ReLU is linear, thus by linearity of the expectation the expected value is preserved. For case (iii), we approximate the ReLU activation function with the affine map $m_k x + t_k$. We first derive a condition to ensure preserving the expected value. After that we show the slope that minimizes the approximation errors is:

$$m_k = \frac{\text{ReLU}(u_{k-1}) - \text{ReLU}(l_{k-1})}{u_{k-1} - l_{k-1}} = \frac{u_{k-1}}{u_{k-1} - l_{k-1}}. \tag{34}$$

The expected value is preserved if the offset $t_k$ to satisfies the condition:

$$
\begin{aligned}
\mathbb{E}_{h_k \sim \mathcal{H}_k}[h_k] &\overset{!}{=} \mathbb{E}[\text{ReLU}(h_{k-1})] \iff & c_k &= \mathbb{E}[\text{ReLU}(h_{k-1})] \\
&\iff & m_k \, c_{k-1} + t_k &= \mathbb{E}[\text{ReLU}(h_{k-1})] \\
&\iff & t_k &= \mathbb{E}[\text{ReLU}(h_{k-1})] - m_k \, c_{k-1},
\end{aligned} \tag{35}
$$

with $h_{k-1} \sim \mathcal{U}(l_{k-1}, u_{k-1})$. Hence, we fix the offset $t_k$ w.r.t. the slope $m_k$. Moreover, to find the optimal slope $m_k$, we compute the expected value $\mathbb{E}[\text{ReLU}(h_{k-1})]$ using the probability density function $f_{\mathcal{H}_k}$ for the distribution of $\text{ReLU}(\mathcal{U}(l_{k-1}, u_{k-1}))$. Therefore, we first compute the

probability mass below $0$ using the cumulative distribution function (Hinton & Ghahramani, 1997; Socci et al., 1997):

$$F_{\mathscr{U}(l_{k-1}, u_{k-1})}(0) = \int_{-\infty}^{0} f_{\mathscr{U}(l_{k-1}, u_{k-1})}(h_{k-1}) \, \mathrm{d}h_{k-1}$$

$$= \int_{l_{k-1}}^{0} \frac{1}{u_{k-1} - l_{k-1}} \, \mathrm{d}h_{k-1} = \frac{-l_{k-1}}{u_{k-1} - l_{k-1}}.$$

The probability mass is concentrated in a peak at zero using the Dirac delta $\delta(x)$ (Au & Tam, 1999). Hence, we obtain for a compactly supported function $h(x)$, with respect to the measure $\delta$ the Lebesgue integral:

$$\delta(x) = \begin{cases} \infty & x = 0, \\ 0 & \text{else}, \end{cases} \qquad \int_{-\infty}^{\infty} h(x)\,\delta(x)\,\mathrm{d}x = h(0).$$

Thus, the probability density function for the post-activation is

$$f_{\mathcal{H}_k}(h_k) = \begin{cases} \frac{1 - l_{k-1}\,\delta(h_k)}{u_{k-1} - l_{k-1}}, & 0 \le h_k < u_{k-1}, \\ 0 & \text{otherwise}. \end{cases}$$

The resulting density is composed of the uniform distribution for the support $h_k > 0$ and the aggregated probability mass for the negative support $h_{k-1}$ in the Dirac spike at $h_k = 0$. This follows immediately from the definition of $\mathrm{ReLU}(h_{k-1})$. Thus, the expected value of the transformed distribution is given by:

$$\mathbb{E}_{h_{k-1} \sim \mathscr{U}(l_{k-1}, u_{k-1})}[\mathrm{ReLU}(h_{k-1})] = \int_{-\infty}^{\infty} \mathrm{ReLU}(h_{k-1})\, f_{\mathcal{H}_k}(\mathrm{ReLU}(h_{k-1}))\, \mathrm{d}h_{k-1}$$

$$= \int_{0}^{u_{k-1}} h_k\, f_{\mathcal{H}_k}(h_k)\, \mathrm{d}h_k$$

$$= \int_{0}^{u_{k-1}} h_k\, \frac{1 - l_{k-1}\,\delta(h_k)}{u_{k-1} - l_{k-1}}\, \mathrm{d}h_k$$

$$= \frac{h_k^2}{2\,(u_{k-1} - l_{k-1})}\bigg|_{0}^{u_{k-1}} = \frac{u_{k-1}^2}{2\,(u_{k-1} - l_{k-1})}.$$

Hence,

$$t_k \overset{(35)}{=} \frac{u_{k-1}^2}{2\,(u_{k-1} - l_{k-1})} - m_k\, c_{k-1} \overset{(6)}{=} \frac{1}{2}\left(\frac{u_{k-1}^2}{(u_{k-1} - l_{k-1})} - m_k\,(u_{k-1} + l_{k-1})\right). \tag{36}$$

We now find the slope $m_k$ that minimizes the approximation errors; by satisfying (35) we ensure preserving the expected value. For concise notation, we abbreviate the approximation error at $x$ with slope $m_k$ by

$$d_x(m_k) := |(m_k\,x + t_k) - \mathrm{ReLU}(x)|. \tag{37}$$

We optimize the slope $m_k$ for minimal approximation errors:

$$\min_{m_k}\, \max_{x \in [l_{k-1}, u_{k-1}]}\, d_x(m_k).$$

With (Koller et al., 2025, Prop. 7), we know that the approximation error are located at $x \in \{l_{k-1}, 0, u_{k-1}\}$ and rewrite:

$$\min_{m_k}\, \max_{x \in \{l_{k-1}, 0, u_{k-1}\}}\, d_x(m_k).$$

From $l_{k-1} < 0 < u_{k-1}$, we can simplify

$$\begin{aligned} d_{l_{k-1}}(m_k) &= |m_k\, l_{k-1} + t_k|, \\ d_0(m_k) &= |t_k|, \\ d_{u_{k-1}}(m_k) &= |m_k\, u_{k-1} + t_k - u_{k-1}|. \end{aligned} \tag{38}$$

Moreover, we know that at least one approximation error is greater 0:

$$d_{l_{k-1}}(m_k) > 0 \vee d_0(m_k) > 0 \vee d_{u_{k-1}}(m_k) > 0.$$

Hence, the optimal slope $m_k$ is located at an intersection of two error functions $d_{x_1}(m_k) = d_{x_2}(m_k) \geq d_{x_3}(m_k)$ with $\{x_1, x_2, x_3\} = \{l_{k-1}, 0, u_{k-1}\}$. Thus, for each intersection we find the optimal slope:

*Case (2c.i):* $d_{l_{k-1}}(m_k) = d_0(m_k)$

$$d_0(m_k) = d_{l_{k-1}}(m_k)$$

$$\overset{(38)}{\Longleftrightarrow} \qquad |t_k| = |m_k\, l_{k-1} + t_k|$$

$$\Longleftrightarrow \qquad t_k^2 = (m_k\, l_{k-1} + t_k)^2$$

$$\Longleftrightarrow \qquad 0 = (m_k\, l_{k-1})^2 + 2\, m_k\, l_{k-1}\, t_k$$

$$\overset{(36)}{\Longleftrightarrow} \qquad 0 = m_k^2 - \frac{u_{k-1}}{u_{k-1} - l_{k-1}}\, m_k$$

$$\Longleftrightarrow \qquad m_k = 0 \vee m_k = \frac{u_{k-1}}{u_{k-1} - l_{k-1}}.$$

*Case (2c.ii):* $d_{l_{k-1}}(m_k) = d_{u_{k-1}}(m_k)$

$$d_{l_{k-1}}(m_k) = d_{u_{k-1}}(m_k)$$

$$\overset{(38)}{\Longleftrightarrow} \qquad |m_k\, l_{k-1} + t_k| = |m_k\, u_{k-1} + t_k - u_{k-1}|$$

$$\Longleftrightarrow \qquad (m_k\, l_{k-1} + t_k)^2 = (m_k\, u_{k-1} + t_k - u_{k-1})^2$$

$$\overset{(36)}{\Longleftrightarrow} \qquad m_k\, u_{k-1}\, l_{k-1} = u_{k-1}\, \frac{u_{k-1}^2}{u_{k-1} - l_{k-1}} - u_{k-1}^2$$

$$\Longleftrightarrow \qquad m_k = \frac{u_{k-1}}{u_{k-1} - l_{k-1}}.$$

*Case (2c.iii):* $d_0(m_k) = d_{u_{k-1}}(m_k)$

$$d_0(m_k) = d_{u_{k-1}}(m_k)$$

$$\overset{(38)}{\Longleftrightarrow} \qquad |t_k| = |m_k\, u_{k-1} + t_k - u_{k-1}|$$

$$\Longleftrightarrow \qquad t_k^2 = (m_k\, u_{k-1} + t_k - u_{k-1})^2$$

$$\overset{(36)}{\Longleftrightarrow} \qquad -m_k^2\, l_{k-1} + m_k\left(\frac{l_{k-1}\,(2\, u_{k-1} - l_{k-1})}{u_{k-1} - l_{k-1}}\right)$$

$$= \frac{u_{k-1}^2}{u_{k-1} - l_{k-1}} + u_{k-1}$$

$$\Longleftrightarrow \qquad m_k = 1 \vee m_k = \frac{u_{k-1}}{u_{k-1} - l_{k-1}}.$$

In each case, we show that the proposed slope in (34) is optimal and minimizes the approximation errors $\underline{d}_k, \overline{d}_k \in \{d_{l_{k-1}}, d_0, d_{u_{k-1}}\}$.

$\square$

## C    DISCLOSURE: USAGE OF LARGE LANGUAGE MODELS (LLMs)

We used a large language model (LLM) as a general-purpose assist tool to aid in polishing the writing and improving clarity of expression. The research ideas, methodology, analysis, and conclusions are the authors' own. The LLM did not contribute to the ideation, design, or execution of the research.

