# OpenReview forum: "Training Verifiably Robust Agents Using Set-Based Reinforcement Learning"
_ICLR.cc/2026/Conference — Submitted to ICLR 2026_

### Official Review · Reviewer_fozE · 2025-10-22

**Soundness:** 4
**Presentation:** 2
**Contribution:** 3
**Rating:** 4
**Confidence:** 4

**Summary:**

Set-based reinforcement learning (RL) is training procedure where policy is optimized to maximize reward under worst-case adversarial perturbation at each step. The method uses zonotope propagation (“set-based” approach) to compute lower bounds for reward, in analogy with certified training used for image classifiers. In the paper, two such algorithms (SA-SC and SA-PC) are presented. They produce policies that are significantly more robust than previous methods based on heuristic adversarial attacks. However, these set-based RL algorithms also demonstrate a noticeable decrease in both natural reward (without perturbations) and adversarial reward (under attacks), as can be seen in Figure 7. The paper further provides theoretical justification for the proposed algorithms.

**Strengths:**

This paper presents strong contribution by introducing new policy training algorithm that maximizes worst-case rewards, which is important for safety-critical systems. The experimental evaluation shows clearly that proposed algorithms outperform existing methods in terms of verified rewards on standard dynamical system benchmarks. The proofs included in appendix are brief but correct, and overall the paper is written in satisfactory manner.

**Weaknesses:**

Despite valuable contribution, this paper has several weaknesses which limit its overall impact. The main issues are following:

1. The term reinforcement learning seems not fully appropriate for described algorithm. The method is designed specifically for control of dynamical systems, and environment setting is never stated explicitly. From examples, it is clear that all benchmarks are deterministic continuous-time systems. Such setup is not typical for reinforcement learning, since it lacks stochasticity which is essential component of RL formulation.

2. The SA-SC algorithm appears unnecessary. Motivation for introducing “exploration noise” in actions (lines 211–214) is unclear. Moreover, experimental results indicate that SA-SC is in most cases outperformed by SA-PC.

3. Section 4 derivation relies on rather arbitrary assumptions about probability distributions. In particular, the claim that expected value over zonotope equals its center is quite restrictive and limits generality of possible distributions.

4. Proposition 4.1 is mathematically correct but its presentation is misleading. More precise statement would be: “There exists a zonotope relaxation of ReLU such that $E_{h_k \sim H_{k}}[h_k] = E_{h_{k-1} \\sim U(\ell_{k-1},u_{k-1})}[L_k(h_{k-1})]$. This zonotope relaxation has $m_k = \ldots$.” The proof holds only for ReLU networks, and this restriction should be written explicitly. Furthermore, the paragraph before Proposition 4.1 creates confusion and even suggests that proposition might be false, probably due to use of different relaxation than the one stated. The proof ends abruptly and does not explain why approximation errors $\underline{d}_k, \overline{d}_k$ are equal to $t_k$, which is necessary for concluding that zonotope center equals expected value of ReLU under uniform input.

5. Figure 7 shows that proposed algorithms achieve lower natural and adversarial rewards, but this observation is not analyzed in text. The same metrics should be reported also for other benchmarks. Certified training methods usually involve trade-off between robustness and natural accuracy, and there can remain significant gap between verified and true worst-case performance. These two aspects require explicit discussion.

6. Colour scheme in Figures 5–11 complicates interpretation. It is difficult to distinguish PA-PC from SA-SC ($\omega=0.5$).

**Questions:**

The authors may provide comments regarding weaknesses mentioned above.

*List of Typos and Additional Remarks*

- Abstract should begin with “Deep,” because statement “Reinforcement learning uses neural networks” is not correct for all RL methods.
- Replay buffer is denoted as $D$ in Algorithm 1, but as $B$ in text, which causes inconsistency.
- In line 783 symbol used for regression loss differs from that in main text.
- Function $f$ in line 828 should be defined.
- Equality in line 835 appears inaccurate; probably density inside integral must more likely be $U(\ell_{k-1}, u_{k-1})$.

---

> ### Author Response · Authors · 2025-11-20
>
> Dear reviewer fozE,
>
> thank you for reviewing our paper and contributing to the helpful discussion. We are thankful for your opinion on the strong contribution of our paper and hope to improve clarity and presentation based on your valuable comments.
>
>
> > Reinforcement learning
>
> Thank you for raising this point. Our method is based on the Deep Deterministic Policy Gradient [1]. It follows the standard RL interaction loop: the agent observes the system state, selects actions via a parameterized policy, and updates this policy from collected transitions. The environment is described in Sec. 2.3, where we formally define the action space, the state space, the dynamics, and the reward, which form a Markov Decision Process.
>
> > Deterministic continuous-time systems
>
> We would like to highlight, to the best of our knowledge, that many among the most important and widely established benchmarks for reinforcement learning study continuous deterministic dynamics [2, 3, 4].
> In this work, we extend several of the tasks from the benchmarks above to a stochastic setting by introducing uncertainty, which is typically present when verifying real safety-critical systems. Further details on the experimental setup can be found in Section 5 and Appendix A.1.
>
>
> > SA-SC algorithm appears unnecessary
>
> We included SA-SC to give a broader picture of our approach.
> Intuitively, by propagating the uncertainty from the value output set back through the critic and the actor, we only penalize the action directions that lead to a larger range of values, while SA-PC in general penalizes any action direction.
> SA-SC has shown advantages for small perturbation radii as shown in Sec. 5.
> Please also refer to our ablation study on the weighting factor $\omega$ for a revised discussion on this topic.
>
>
> > Motivation for exploration noise in actions
>
> Several works show that exploration is required in deep off-policy reinforcement learning algorithms to encourage sufficient exploration [5, 6]. Our approach can naturally include this exploration noise by adding it to the respective action set.
>
>
> > Connection to probability distributions
>
> Thank you for raising this question.
> We are interested in improving the guaranteed worst-case reward in the presence of uncertainty.
> As we do not know which point realizes the worst-case reward,
> we assume a uniform distribution over all points in the set,
> for which the expectation is trivially the center.
> We have found it interesting that with this basic assumption, we can naturally derive our loss function, which intuitively moves the center towards the target for accuracy and shrinks the set for robustness (Fig. 3).
>
>
> > Misleading presentation of Prop. 4.1
>
> Thank you for pointing this out; we have improved the clarity of Prop. 4.1 and addressed the questions regarding the proof in our revised version.
>
> > Rewards for adversarial attacks
>
> Thank you for pointing this out. We have improved and extended our discussion on the performance of SA-PC and SA-SC under different adversarial attacks (Figure 7) in lines 477-487.
>
> Thank you again for thoroughly reviewing our paper. We appreciate your valuable feedback and hope that our response and changes address your questions. We are glad to provide any additional clarification that may support a revised assessment.
>
> ---
> [1] Lillicrap et al. "Continuous control with deep reinforcement learning." ICLR. 2015.
>
> [2] Yuan et al. "Safe-control-gym: A unified benchmark suite for safe learning-based control and reinforcement learning in robotics." IEEE Robotics and Automation Letters 7.4. 2022.
>
> [3] Brockman et al. "Openai gym." arXiv. 2016.
>
> [4] Tassa et al. "Deepmind control suite." arXiv. 2018.
>
> [5] Eberhard et al. "Pink noise is all you need: Colored noise exploration in deep reinforcement learning." ICLR. 2023.
>
> [6] Hollenstein et al. "Action noise in off-policy deep reinforcement learning: Impact on exploration and performance." arXiv. 2022.

---

> > ### Comment · Reviewer_fozE · 2025-11-23
> >
> > The authors have made several good changes following my suggestions. I am not convinced about point 2,3,6. For point 1 paper was unclear for assumptions on transition kernel, which I mistook to be deterministic or with bounded noise. Actually states are perturbed by adversary after environment update.
> >
> > I can see worst-case rewards but unknown distribution does not mean we can assume uniform distribution. True distribution may have expected value away from zonotope center.
> >
> > Overall, rebuttal has increased my evaluation of the paper but I remain critical on several points. Therefore, I changed evaluation to marginal accept.

---

### Official Review · Reviewer_McqD · 2025-10-29

**Soundness:** 3
**Presentation:** 3
**Contribution:** 4
**Rating:** 6
**Confidence:** 3

**Summary:**

This paper introduces set-based reinforcement learning for training verifiably robust agents in continuous control tasks. Unlike adversarial training methods, it uses gradient sets computed from entire input perturbation sets rather than single adversarial examples. The approach enables formal verification for up to 9 times larger input perturbations across different verification frameworks.

**Strengths:**

Significant novel contribution introducing first set-based RL algorithm bridging adversarial training and formal verification. Major theoretical innovation using gradient sets from entire perturbation sets. Results demonstrate up to 9x larger perturbation tolerance compared to state-of-the-art methods while maintaining formal verifiability across multiple verification frameworks (CORA, CROWN-Reach, JuliaReach, NVV). The paper tests across multiple established benchmarks (Navigation Task, 1D/2D Quadrotor, Inverted Pendulum) with proper statistical analysis, ablation studies, and comparisons against multiple baseline methods including recent adversarial training approaches.

**Weaknesses:**

The Navigation Task incorporates safety through reward penalties rather than explicit constraints during training. The authors acknowledge that formal safety guarantees require separate verification beyond just reward lower bounds, limiting the direct safety claims.

Evaluation focuses primarily on continuous control tasks with relatively low-dimensional state/action spaces. The approach's effectiveness on high-dimensional problems (e.g., image-based RL) or discrete action spaces remains unclear. However, this is a common limitation in this area across majority of approaches.
The method introduces several new hyperparameters that require careful tuning. The paper provides limited guidance on hyperparameter selection across different domains, and the sensitivity analysis is incomplete, potentially limiting practical adoption.

**Questions:**

How does the computational complexity and memory usage of your set-based training compare to a,b-CROWN's verification approach? Can you provide detailed runtime comparisons on larger networks and discuss the trade-offs between training-time robustness vs. after-training verification?
Does the approach scale to high-dimensional observation spaces (e.g., image-based RL tasks)? How is the scalability compared to ab-crown?


The current work focuses on continuous control; can the set-based approach be extended to discrete action spaces?

---

> ### Author Response · Authors · 2025-11-20
>
> Dear Reviewer McqD,
>
> thank you very much for your time and effort in reviewing our paper. We appreciate your view on the novelty and contribution of our work. We are thankful for the valuable comments and happy to improve our paper based on these.
>
>
> > Hyperparameters discussion.
>
> To address this useful question, we add a short discussion and two additional experiments in Appendix A.3 regarding the influence of the introduced hyperparameters. We also provide information about the convergence behavior of set-based reinforcement learning across the different hyper parameters.
>
>
> > Runtime complexity and after-training verification
>
> Based on your comment, we have added a discussion on the runtime complexity of our approach in Appendix A.4.
> The advantage of the used zonotope-based method over $\alpha,\beta$-Crown is that zonotopes are an abstract representation of entire sets, over which we can explicitly define sets of gradients. Whereas $\alpha,\beta$-Crown is based on MILPs to derive individual inequalities for the output, which limits the usage of explicit gradients with respect to the neural network parameters over the entire set.
>
> However, after training, any verification method can be used and we have seen benefits of agents trained using our approach for all verification approaches (Tab. 1).
> We discuss this in more detail in the revised version.
>
>
> > Discrete action spaces
>
> Thank you for your question on extending the methodology to discrete action spaces. We focused on continuous state and action spaces because many real-world safety-critical problems naturally fall into this domain. For an extension of our learning algorithm to discrete action spaces, one is required to discretize the computed action set and compute the gradient with respect to that discretization. One path could be to use the discretization approach described in [1], which verifies a continuous dynamical system with discrete control inputs obtained via decision trees. There, zonotopes are propagated through the nodes of the decision tree to obtain a discrete action. However, they do not consider robust training or compute any gradients. We include a full discussion on this topic in the revised version.
>
> Thank you once again for your thorough review. We hope the additional explanations and revisions are helpful in addressing your comments, and we would be grateful for your consideration of our revisions in updating your assessment.
>
> ---
> [1] Schilling et al. "Safety Verification of Decision-Tree Policies in Continuous Time". NeurIPS. 2023.

---

### Official Review · Reviewer_nxrK · 2025-11-01

**Soundness:** 3
**Presentation:** 3
**Contribution:** 2
**Rating:** 4
**Confidence:** 3

**Summary:**

This paper introduces a **set-based reinforcement learning (RL)** framework that extends set-based neural network training to the actor–critic setting. The core idea is to **propagate uncertainty sets (zonotopes)** through the policy and value networks to obtain bounds on outputs and gradients. The method defines a **set-based regression loss** for the critic that penalizes both prediction error and set diameter, and a **set-based policy gradient** for the actor that encourages compact output sets. The approach aims to yield RL policies that are **verifiably robust** under input perturbations. Experimental results on navigation and control benchmarks demonstrate improved verified performance and robustness compared to adversarially trained baselines.

**Strengths:**

**Originality**
- The work is an interesting and timely attempt to unify **formal verification and reinforcement learning**, using set-based propagation within the training loop.
- The idea of **gradient sets** and integrating them with actor–critic architectures is novel and potentially impactful.

**Quality**
- The theoretical exposition (e.g., use of zonotopes, set propagation through affine and activation layers) is mostly sound.
- The proposed framework connects well to existing verification methods, showing that trained policies can be more amenable to reachability-based certification.

**Clarity**
- The algorithm is clearly presented, and the basic zonotope operations are described in a straightforward way.
- The empirical section reports verified robustness metrics under multiple verification toolboxes, which adds credibility.

**Significance**
- The topic—training verifiably robust RL agents—is important and underexplored. The framework could stimulate further research into **hybrid learning–verification paradigms**.

**Weaknesses:**

1. **Over-approximation accumulation**
   - The framework propagates set over-approximations step-by-step in an RL setting, but there is **no theoretical bound** on how the set diameter grows over time. Without such analysis, the reachability sets may become **too conservative** (i.e., trivial or uninformative), limiting practical use.

2. **Goal reachability and certification**
   - The paper uses over-approximated reachable sets to argue about goal attainment, but **overlap with the goal set does not imply actual reachability**. There is no mechanism ensuring that the system trajectory truly enters the goal region.

3. **Complexity and scalability**
   - The authors do not analyze the **computational complexity** of set propagation or describe how generator matrices are pruned to maintain tractability. The absence of any **scalability discussion** (time/memory vs. state dimension) is concerning.

4. **Notation and clarity issues**
   - The same symbol \(\theta\) appears at different lines (e.g., 105 vs. 124) for possibly different objects—this causes confusion.
   - In Eq. (11), the variable \(t\) appears in \(\nu(s_t, t)\) but is not used later; this seems meaningless unless time-varying noise is considered.

5. **Activation coverage and approximation errors**
   - Proposition 4.1 only discusses ReLU networks. The **approximation error bounds** for non-ReLU activations are not analyzed.

6. **Reward design justification**
   - The reward function used at line 418 lacks motivation or theoretical connection to the claimed “verified performance.” The impact of reward shaping on the certification outcome is not studied.

7. **Runtime and scalability reporting**
   - The paper omits **training runtime**, **state/action dimensions**, and **network sizes**. Table 1 shows only verification time, making it difficult to evaluate the **overall efficiency**.

**Questions:**

1. **Over-approximation accumulation**
   - How do you ensure that the reachable sets remain non-trivial as uncertainties propagate across time?
   - Is there a formal bound on the diameter growth of the sets, e.g., as a function of the Lipschitz constants of dynamics and policy?

2. **Propagation until termination**
   - Does “termination” refer to a fixed time horizon or a goal-reaching condition?
   - How is the reachability propagation stopped, and how is goal satisfaction formally defined?

3. **Goal set overlap vs. actual reachability**
   - Since the approach relies on outer approximations, how can we be sure the agent truly **reaches** the goal set rather than merely overlapping with it?

4. **Parameter notation**
   - Are the two symbols \(\theta\) (line 105 and line 124) referring to the same network parameters? If not, please rename one (e.g., \(\vartheta\) or \(\Theta\)).

5. **Equation (11)**
   - What is the purpose of \(t\) in \(\nu(s_t, t)\)? If it is redundant, it should be removed for clarity.

6. **Equation (12)**
   - Please clarify what \(c\) and \(G\) represent and how they are derived from the actor network’s output enclosure. Are they the zonotope center and generator matrix computed via affine propagation?

7. **Proposition 4.1**
   - How does the result generalize to **non-ReLU activations** such as \(\tanh\)?
   - What are the quantitative approximation errors introduced by these nonlinearities?

8. **Complexity and computation**
   - What is the time complexity of a single forward pass and backward pass in your set-based framework?
   - Do you apply any **order-reduction heuristics** to control the explosion of zonotope generators?

9. **Reward design**
   - Why was the specific reward function in line 418 chosen?
   - Does it have any special properties that make verification easier or bounds tighter?

10. **Runtime and scalability**
    - Please report wall-clock training time, state/action dimensions, and network sizes for all tasks.
    - How well does the method scale to high-dimensional or long-horizon RL problems?

**Details Of Ethics Concerns:**

No ethics concerns.

---

> ### Author Response · Authors · 2025-11-20
>
> Dear Reviewer nxrK,
>
> thank you for reviewing our paper and highlighting the potential and significance of the research direction.
>
>
> > Over-approximation accumulation
>
> The reviewer is right in their assessment that verification based on reachability analysis usually does not provide theoretical bounds of the diameter growth over the entire state space.
> However, in practice, one can compute the reachable set for a continuous time horizon and only proceed if this time horizon can be verified. Otherwise, a fail-safe operation is executed, which was computed in the previous (already verified) verification step, with the induction starting with the agent, e.g., an autonomous car, standing still.
> In Tab. 1, we show that using our set-based training approach, CORA can enable the verification of a time horizon of $8s$ in $<2s$, demonstrating a clear improvement over previous approaches that require $>100s$.
> Please also note that CORA is written in MATLAB, and additional speed-ups can be achieved by implementing it in more performant programming language like C/C++.
>
>
> > Goal reachability and certification
>
> It is true that an overlap does not guarantee to successfully reach the goal.
> However, please note that the computed reachable set contains more information than one can show in a static image; specifically, the final reachable set consists of a union of reachable sets for subsequent time intervals.
> Thus, we can check if the reachable set in the final time interval is fully contained in the goal set.
> We can confirm that this is the case for our set-based training approach.
> You can also see this in the video we provide: https://t1p.de/zjayi,
> where the individual time intervals are plotted sequentially over time.
>
>
> > Complexity and scalability
>
> Thank you for this suggestion.
> We have added a discussion on the runtime complexity and scalability in Appendix A.4.
>
>
> > Clarity issue: In Eq. (11), the variable t appears in $\nu(s_t, t)$ but is not used later.
>
> Thank you for pointing this out. We would like to clarify that the time $t$ in $\nu(s_t,t)$ indicates that the perturbation is not stationary but can essentially differ when revisiting the state.
>
>
> > Activation coverage and approximation errors
>
> We are happy to point out that our approach is also applicable for non-ReLU networks. We clarified that we use the derived enclosure by Koller et. al. [1] for all activations (Eq. 9). This work discusses the tightness of the provided outer approximations and compares it against the established work by Singh et. al.
> Our goal is to take advantage of existing works [1, 2] for set-based reinforcement learning and establish a principled derivation for the set-based regression loss and the set-based policy gradient in sections 4.1 to 4.3.
>
>
> > Reward design
>
> Our set-based reinforcement learning algorithm is agnostic to the reward function, and the $\ell_1$ reward is chosen considering robustness for learning tasks in the presence of perturbations [3], see our discussion with reviewer jELz.
>
>
> > Runtime and hyperparameters
>
> Thank you for mentioning this. We would like to point you to Table 3 in Appendix A.4 for the different training time comparisons. Appendix A.1 contains detailed information about the individual benchmarks, including state and action dimensions. Further, Table 2 in Appendix A.2 provides an extensive list of all hyperparameters used during the experiments.
>
>
> > Termination
>
> It is a typical practice in reinforcement learning to truncate infinite-horizon tasks after a fixed finite number of steps [4, 5].
> Verification is then determined as stated above in our goal reachability answer.
>
>
> > Clarification of notation
>
> We would like to highlight that the center of the zonotope $c$ and the generators $G$ are defined in Definition 2.2. We leverage the set-based computations introduced in Section 2.4, to propagate the Zonotopes through the individual layers (Proposition 2.3, Equations 8 and 9).
> We also clarified all other identifed issues in the revised version.
>
>
> > Zonotope order reduction
>
> We currently do not apply order reduction techniques during the training process.
> Please refer to the discussion with reviewer jELz on how this could be realized.
>
> Thank you again for your review and feedback. We hope our revisions address your concerns, and would appreciate your consideration of a score update. We are of course happy to clarify any further question.
>
> ---
> [1] Koller et al. "Set-based training for neural network verification." TMLR. 2025.
>
> [2] Singh et al. "Fast and effective robustness certification." NeurIPS. 2018.
>
> [3] Huber, Peter J. "Robust estimation of a location parameter." Annals of Mathematical Statistics. 1964.
>
> [4] Puterman, Martin L. "Markov decision processes: discrete stochastic dynamic programming". John Wiley and Sons. 2014.
>
> [5] Sutton et al. "Reinforcement learning: An introduction". Vol. 1. No. 1. Cambridge: MIT press. 1998.

---

### Official Review · Reviewer_jELz · 2025-11-01

**Soundness:** 3
**Presentation:** 3
**Contribution:** 2
**Rating:** 4
**Confidence:** 4

**Summary:**

This paper proposes a **set-based reinforcement learning (RL)** framework that integrates *set-based neural network training* with the *actor–critic paradigm* to achieve **formally verifiable robustness**. The method introduces *gradient sets* to account for uncertainty propagation through neural networks: each possible output under input perturbations has a corresponding gradient. By minimizing the size of propagated sets during training, the resulting policy and value functions exhibit improved *worst-case guarantees* and can be verified using standard reachability tools such as CORA. The experiments across benchmarks (e.g., 1D/2D Quadrotor, Inverted Pendulum, Navigation Task) demonstrate improved verified performance and robustness compared to adversarially trained agents, extending recent work on set-based training for feed-forward networks.

**Strengths:**

- The work extends **set-based neural network training** to the reinforcement learning setting, introducing gradient sets into both actor and critic updates.
- It offers an elegant synthesis of *verification-oriented training* and *policy optimization*, bridging robust RL and formal methods.
- The proposed set-based regression loss and policy gradient are mathematically grounded in probabilistic reasoning, linking the *likelihood–prior decomposition* with set diameter minimization.
- The results showing successful verification across multiple tools (CORA, CROWN-Reach, JuliaReach, NNV) indicate cross-framework generality.
- The structure of the paper is clear and self-contained, with strong visualizations (e.g., Figs. 1, 3, 5–7) and detailed pseudocode (Algorithm 1).

**Weaknesses:**

1. **Theoretical limitations of over-approximation**
   - The approach relies on *outer approximations* of reachable sets, yet the paper provides no quantitative analysis of **set overgrowth or error bounds**.
   - This makes it unclear whether the final reachability sets meaningfully constrain the true behavior, especially over long horizons.

2. **Lack of computational analysis**
   - While the framework is elegant, there is no complexity or runtime discussion of *set propagation* or *gradient set computation*.
   - The claim of scalability (e.g., to 2D Quadrotor and Hopper-v2) is qualitative and unsupported by profiling data.

3. **Empirical evidence limited in scope**
   - The experiments demonstrate robustness but not **training stability** or **verification time trade-offs** in detail.
   - It remains unclear whether set-based training slows convergence compared to standard or adversarially trained DDPG.

**Questions:**

1. **On over-approximation accumulation**
   - How does the set diameter evolve across time steps?
   - Do you apply zonotope order reduction or pruning to prevent explosion of generator dimensions?

2. **Verification tightness**
   - Can the authors quantify the difference between the *verified lower bound* and the *empirical return*?
   - What percentage of trajectories are “verified tight” (i.e., within a small margin)?

3. **Scalability**
   - What is the computational cost of one training epoch compared to adversarial training (e.g., FGSM-based)?
   - How does verification time grow with network depth or state dimension?

4. **Reward function and verification metric**
   - Why was $r(s,a)=w^\top|s-s^*|$ chosen? Would other smooth reward forms (e.g., quadratic penalties) affect the verified performance definition in Eq. (28)?

5. **Theoretical extension**
   - Could the framework be adapted for stochastic policies or discrete action spaces?
   - Is there an underlying connection between your set-based gradient and **Lipschitz regularization** approaches?

6. **Practicality**
   - Can the proposed method handle **nonlinear system models** beyond the evaluated tasks (e.g., with unmodeled dynamics or contact)?
   - Have you observed instability due to conservative set updates in high-dimensional tasks?

---

> ### Author Response · Authors · 2025-11-20
>
> Dear Reviewer jELz,
>
> thank you very much for your time and valuable comments. Please find our detailed responses below.
>
>
> > Theoretical limitations of over approximation
>
> The reviewer is right in their assessment that verification based on reachability analysis usually does not provide theoretical bounds of the diameter growth over the entire state space.
> Please see also our discussion with reviewer nxrK about reachability analysis and goal reachability.
> However, our approach substantially reduces the outer approximation compared to standard training, even for large perturbations (Fig. 5 and 8),
> thus, improving formal verifiability.
> This result also generalizes to other toolboxes (Tab. 1), reducing the verification time from 130s to just 2s.
>
> To ensure safety over long time horizons, e.g., showing stability as in the NAV example (Fig. 1), one can check if the computed reachable set of a discretization step is contained within the previous step [1], which guarantees safety for an infinite time horizon.
> Please note that this then holds for the entire continuous time, not just at the discretization steps.
> This enables the safe deployment of set-based trained agents in safety-critical environments.
>
>
> > Lack of computational analysis
>
> To address more insights in the runtime of our algorithm, we have added a discussion in Appendix A.4, where we formally state the time complexity of our algorithm.
>
>
> > Does set-based training slow convergence?
>
> We can confirm that we have not observed a different convergence or learning behavior compared to vanilla DDPG or adversarial methods. We have added a discussion about this in Appendix A.3.
>
>
> > Diameter evolvement over time
>
> This heavily depends on the stability of the controller for a given perturbation and initial conditions.
> An example with a large perturbation radius can be seen in Fig. 6, where the diameter for the SA-PC (light blue) agent stays bounded over the entire time horizon.
>
>
> > Zonotope order reduction
>
> This is indeed an interesting question. Currently, we do not include order reduction techniques during training.
> However, it would be valuable, since it would speed up the training process and provide scalability to larger architectures.
> Please note that this requires a gradient computation through the order reduction operation.
> This can be done for simple order reductions as a box enclosure (as it is just based on sums of absolute values of the involved generators), but might be trickier for more complex order reduction techniques.
> Thanks again for this suggestion, we will definitely consider this in our ongoing research on this topic!
>
>
> > Verified lower bound vs. empirical return
>
> Please note that we report in Figure 7 both the verified performance of SA-PC, SA-SC, and the baseline methods, as well as their empirical performance under two different adversarial attacks. As expected, the verified performance decreases with increasing perturbation radius $\\epsilon_{\\text{test}}$ and consistently lies below the empirical performance under attack. Quantitatively, the difference between the attack-based empirical upper bound of the worst-case performance and the formally verified lower bound grows with $\\epsilon_{\\text{test}}$.
> Below, we list the average values for $\\epsilon_{\\text{test}}=0.1$.
> However, the empirical return is not sufficient for safety-critical domains.
> Thus, the verified lower bounds have to be considered, for which agents trained with our approach perform substantially better.
>
> | Algorithm | LBs | Naive | Grad |
> |-----------|-----|-------|------|
> |PA-PC|-198|-27.4|-26.8|
> |Naive|-187|-25.9|-25.8|
> |Grad|-215|-26.0|-26.4|
> |MAD|-165|-25.4|-25.7|
> |SA-PC|-41|-29.2|-29.7|
> |SA-SC $\\omega=0$|-107|-25.0|-25.3|
> |SA-SC $\\omega=0.5$|-28|-25.5|-26.0|
>
>
>
> > Computational cost of one training epoch
>
> We added an analysis of the computational complexity in Appendix A.4.
> Our method is, as expected from this complexity, not significantly slower compared to the adversarial baselines (Table 3), but is the only method that can be verified effectively by state-of-the-art verification toolboxes (Table 1).
>
>
> > Reward function
>
> Thank you for pointing this out. Generally speaking, the choice of the reward can have a significant effect on non-robust methods. For instance, a quadratic reward is more sensitive to large perturbations, compared to $\ell_1$. Therefore, to make the experiments feasible to the non-robust baselines, we choose the $\ell_1$ reward structure, motivated by the Huber Loss [2].
> However, our set-based training algorithm is agnostic to the reward and other rewards can be chosen as well.

---

> > ### Author Response · Authors · 2025-11-20
> >
> > > Stochastic policies
> >
> > Thank you for this question. In principle, one could extend our algorithm to stochastic policies. This could be done by propagating the action mean and the variance as zonotopes. However, we would like to highlight that actions produced by stochastic policies are counteractive for the desired deterministic verification of the controller under input perturbations.
> >
> >
> > > Discrete actions
> >
> > We focus on continuous state and action spaces because this is applicable to many real-world safety-critical applications. For an extension to discrete action spaces, one is required to discretize the computed action set and compute the gradient with respect to that discretization. One path could be to use the discretization approach described in [3]. For a more thorough elaboration we kindly refer you to our discussion with reviewer McqD.
> >
> >
> > > High-dimensional and nonlinear system models
> >
> > Thank you for bringing this point up. As shown in Appendix A.3, we can apply the learning algorithm to highly nonlinear dynamical systems. A model is for all benchmarks only required for formal verification after training. We demonstrate in Appendix A.3 that our algorithm significantly increases the empirical robustness also on these high-dimensional tasks with contact forces, for which, to the best of our knowledge, no formal verification toolbox is available yet.
> >
> >
> > We hope our revisions address your comments, and we would appreciate your consideration of a score update. We are of course happy to clarify any further questions. Thank you again for your review and feedback.
> >
> > ---
> > [1] Althoff, Matthias. "Reachability analysis and its application to the safety assessment of autonomous cars." PhD Thesis. 2010.
> >
> > [2] Huber, Peter J. "Robust estimation of a location parameter." Annals of Mathematical Statistics. 1964.
> >
> > [3] Schilling et al. "Safety Verification of Decision-Tree Policies in Continuous Time". NeurIPS. 2023.

---

> > ### Comment · Reviewer_jELz · 2025-11-23
> >
> > Can you briefly summarize the reason why the outer overapprximation is significantly reduced by your approach and why it is much more efficient, from 130s to 2s?

---

> > > ### Author Response · Authors · 2025-11-23
> > >
> > > The verification time is decreased as agents trained with our approach have seen perturbations during training, and got their weights adjusted to minimize the accumulated approximation error during verification due to these perturbations (robustness term in Prop 3.1 and Def. 3.2, visually in Fig. 3). This allows us to verify the system using the entire initial set in one verification run, whereas standard trained agents require expensive branch-and-bound techniques to verify the system (Tab. 1). Please note that agents trained with adversarial attacks do not bring these benefits (Fig. 7).

---

### Author Response · Authors · 2025-11-29

Dear AC,

Thank you for taking over the area chair for our paper.
We found the feedback provided by the reviewers very thoughtful and detailed, and the constructive discussion period led to a substantially improved quality of our paper. This was also acknowledged by the reviewers with a score update.
Details on our main changes are summarized below:

**Learning History and Hyperparameter Influence**
To address concerns about potential instability or brittleness during training, we included learning-history plots that demonstrate stable and consistent learning behavior throughout the training process.

In response to reviewer McqD’s suggestion, we added a detailed discussion of the hyperparameter influence to enhance practical usability.

We added all missing ablation studies in Appendix A.3.

**Computational Complexity**
Reviewers also noted the absence of a formal discussion on computational complexity. In response, we added a dedicated section that provides a formal computational-complexity analysis in Appendix A.4, complementing our empirical results.

**Issues with Proposition 4.1**
Following reviewer fozE’s review, we improved the phrasing of Proposition 4.1 and its accompanying proof. These revisions clarify the underlying assumptions and prevent possible misinterpretations.

**Additional clafirifcations and Other Revisions**
Moreover, we revised the entire paper based on the remaining feedback of the reviewers, including open points due to the early end of the discussion period, such as the question about the improved verifiability by reviewer jELz, and adjusting the color scheme for clarity.

Thank you again for all the valuable feedback that substantially improved our paper.

The Authors.

---

### Meta-Review · Area_Chair_uhzg · 2025-12-18

**Summary:**

This paper proposes a novel and theoretically elegant set-based reinforcement learning framework to train verifiably robust policies. While reviewers acknowledge its originality in bridging robust RL and formal verification, and its empirical results across multiple verification tools, fundamental methodological concerns necessitate rejection. The core weaknesses are theoretical and practical: the framework lacks analysis of over-approximation error bounds and set diameter growth over time, making its verification guarantees potentially unsound or overly conservative. Furthermore, the paper omits critical computational complexity analysis, scalability profiling, and comparisons to standard robust training baselines in terms of convergence stability.  Though in the rebuttal and the revised draft, the authors have partially addressed some of these concerns, the work still does not look technically solid and sound. Some unresolved issues highlighted consistently across multiple reviews prevent the work from making a complete and actionable contribution in its current form, despite its promising conceptual direction.

**Reviewer Concerns:**

Comments from Reviewers nxrK and jELz are still somewhat outstanding, though the comments from the other two reviewers were mostly addressed.

**Reviewer Scores:**

Reviewer fozE will increase the score. However, the rest of reviewers won't increase the scores.

---

### Decision · Program_Chairs · 2026-01-26

Reject